# Massively parallel in vivo CRISPR screening identifies RNF20/40 as epigenetic regulators of cardiomyocyte maturation

Nathan J. VanDusen [1], Julianna Y. Lee[1], Weiliang Gu[1,2], Catalina E. Butler [1], Isha Sethi[1], Yanjiang Zheng[1,3], Justin S. King [1], Pingzhu Zhou[1], Shengbao Suo[4], Yuxuan Guo [1], Qing Ma[1], Guo-Cheng Yuan[5] & William T. Pu[1,6 ✉]

The forward genetic screen is a powerful, unbiased method to gain insights into biological processes, yet this approach has infrequently been used in vivo in mammals because of high resource demands. Here, we use in vivo somatic Cas9 mutagenesis to perform an in vivo forward genetic screen in mice to identify regulators of cardiomyocyte (CM) maturation, the coordinated changes in phenotype and gene expression that occur in neonatal CMs. We discover and validate a number of transcriptional regulators of this process. Among these are RNF20 and RNF40, which form a complex that monoubiquitinates H2B on lysine 120. Mechanistic studies indicate that this epigenetic mark controls dynamic changes in gene expression required for CM maturation. These insights into CM maturation will inform efforts in cardiac regenerative medicine. More broadly, our approach will enable unbiased forward genetics across mammalian organ systems.

---

[1] Department of Cardiology, Boston Children's Hospital, Boston, MA, USA. [2] Department of Pharmacology, School of Pharmacy, Shanghai University of Traditional Chinese Medicine, Shanghai, China. [3] Department of Biochemistry, West China School of Basic Medical Sciences & Forensic Medicine, Sichuan University, Chengdu, Sichuan, China. [4] Department of Pediatric Oncology, Dana-Farber Cancer Institute, Boston, MA, USA. [5] Department of Genetics and Genomic Sciences and Institute for Personalized Medicine, Icahn School of Medicine at Mount Sinai, New York, NY, USA. [6] Harvard Stem Cell Institute, Cambridge, MA, USA. ✉email: william.pu@cardio.chboston.org

At birth, mammalian CMs undergo maturation, a dramatic and coordinated set of structural, metabolic, and gene expression changes that enable them to sustain billions of cycles of forceful contraction during postnatal life[1,2]. Fetal CMs are glycolytic, mononuclear, mitotic, and contract against low resistance, whereas adult CMs rely on oxidative phosphorylation, are diploid and postmitotic, and support heart growth by increasing in size (maturational hypertrophy). Sarcomeric and ultrastructural adaptations, such as plasma membrane invaginations known as T-tubules, facilitate coordinated and forceful CM contraction against high resistance. Unfortunately, CMs induced from stem cells or other nonmyocyte sources resemble fetal CMs and lack the hallmark features of mature, adult CMs[3,4]. This "maturation bottleneck" remains a major barrier to using stem cell-derived CMs for disease modeling or therapeutic cardiac regeneration.

The regulatory mechanisms that govern the diverse facets of CM maturation are poorly understood, in large part due to the lack of a suitable in vitro model and challenges associated with in vivo approaches. Although mosaic gene manipulation strategies have allowed more precise interpretation of in vivo experiments with respect to the regulation of maturation[5–8], the low throughput of standard in vivo approaches remains a major barrier. To overcome this obstacle, we sought to perform an in vivo forward genetic screen in mice. The resource intensity of traditional forward genetics has precluded their widespread use in mammals, but Cas9 mutagenesis directed by a library of guide RNAs (gRNAs) makes introduction and recovery of gene mutations highly efficient[9–11], and can be expeditiously deployed in mammals in vivo[5,12,13]. This capability has been used for forward genetic screens in cultured cells[11,14,15], but its ability to interrogate endogenous biological processes in mammals in vivo has yet to be fully realized.

Here we report development of an in vivo forward genetic screen to discover transcriptional and epigenetic regulators of CM maturation. Our screen identified several candidate regulators, including RNF20 and RNF40, genes implicated in congenital heart disease[16,17] that form an enzyme complex that mono-ubiquinates histone H2Bon lysine 120[18].

## Results

**Design of in vivo CRISPR screen for CM maturation factors.** We developed an in vivo forward genetic screen to discover factors that regulate murine CM maturation (Fig. 1a). We employed CRISPR/Cas9 AAV9 (CASAAV)-based somatic mutagenesis[5] and a gRNA library targeting murine transcriptional regulators to create thousands of distinct mutations within different CMs of a single mammalian heart. We screened these mutant CMs with a flow cytometry-based single-cell assay of CM maturation. Sequencing of gRNAs from immature CMs compared with the input library identified gRNAs enriched or depleted in immature CMs, i.e., gRNAs that target genes that cells autonomously promote or antagonize CM maturation, respectively.

To separate individual CMs with immature and mature phenotypes, we took advantage of developmentally regulated sarcomere gene isoform switching, a hallmark of CM maturation[1,19]. In mouse, sarcomere maturation, an essential driver of the overall CM maturation program[7], involves silencing of myosin heavy chain 7 (*Myh7*) and upregulation of *Myh6*. In *Myh7*[YFP] mice[20], YFP is fused to endogenous MYH7, such that YFP fluorescence is controlled by endogenous *Myh7* regulatory elements. We hypothesized that *Myh7*[YFP] could be used as a single-cell readout of CM-maturation state. Examination of *Myh7*[YFP] myocardium from neonatal and adult mice verified that neonatal CMs exhibited strong YFP fluorescence, whereas

YFP was nearly completely silenced in adult CMs (Fig. 1b). Since mutation of both *Gata4* and *Gata6* impairs CM maturation (Suppl. Fig. 1, Suppl. Data 1, and ref. [21]), we next tested the effect of *Gata4/Gata6* mutation on *Myh7*[YFP] expression. We found that CASAAV-Gata4/6, which expresses CM specific Cre and previously validated gRNAs targeting *Gata4* and *Gata6*[7], markedly increased the fraction of transduced CMs that expressed YFP from 11% to 86% (Fig. 1c). These data show that *Gata4/6* mutation inhibits normal CM maturation, including maturational *Myh7* silencing, and supported the use of *Myh7*[YFP] as a single-cell reporter of cardiomyocyte-maturation state.

Because the coordination of CM maturation suggested transcriptional regulation[2], we focused our screen on transcriptional regulators. We developed a pooled CASAAV library comprising AAV-expressing CM-specific Cre and a gRNA designed to target a candidate gene. The candidate gene list (Suppl. Data 2) contained 1894 transcriptional regulators (transcription factors and epigenetic modifiers). Given that GATA4/6 regulate multiple aspects of CM maturation (Suppl. Fig. 1), we also included 259 genes differentially expressed in P6 Gata4/6 high-dose double-KO CMs, and 291 genes with strong adjacent G4 binding at P0 (Suppl. Data 1, 2). In total, 2444 genes were selected for targeting. Using a computational pipeline designed to optimize gRNA on-target activity and yield of frameshift mutations[22,23], we designed six guides for each gene. Seven gRNAs that do not target the mouse genome were also included as negative controls. These 14675 gRNAs were synthesized as an oligonucleotide pool, cloned into the CM specific CASAAV vector, and packaged into AAV9 (see "Methods"). To introduce a positive control, a small amount of the CASAAV-Gata4/6 vector was spiked into the AAV pool.

**Execution of in vivo CRISPR screen for CM-maturation factors.** We subcutaneously injected the AAV library into newborn *R26*[Cas9-GFP/+];*Myh7*[YFP/+] pups at a dose sufficient to transduce approximately 50% of the myocardium. CMs were isolated from mice at four weeks of age. YFP⁺ CMs were isolated from the input pool by flow cytometry (Suppl. Fig. 2a). gRNAs in YFP⁺ and input pools were quantified by next generation sequencing. We analyzed 45 hearts in pools of three, resulting in 15 input and 15 YFP⁺ samples. In total, four million transduced CMs were analyzed, of which 365,000 were YFP⁺. In total, 11 YFP⁺ and 14 input samples passed quality control (Suppl. Fig. 2) and separated into distinct clusters (Fig. 1d). In total, 834 gRNAs were significantly enriched within YFP⁺ samples (adj. $P < 0.001$; Suppl. Data 3), with the *Gata4* and *Gata6* positive control gRNAs being among the most enriched (Fig. 1e). The seven human-targeting negative control gRNAs did not show enrichment (Fig. 1f). We used MaGeCK[24] to consolidate the six gRNA enrichment scores per gene into a single score. gRNAs targeting 121 genes were significantly enriched within YFP⁺ CMs, while gRNAs targeting 148 genes were depleted ($P < 0.05$; Suppl. Data 3). The top-ranked depleted gene was *Myh7* (Fig. 1g), which was expected given that YFP fused to *Myh7* was the screen readout. Among the enriched genes were thyroid hormone receptor alpha (*Thra*) and nucleolin (*Ncl*), established regulators of maturation[25,26], as well as many novel candidates (Fig. 1h; Suppl. Data 3), including both *Rnf20* and *Rnf40*, which encode components of an epigenetic complex that monoubiquitinates histone 2B[27,28].

**Validation of candidate factors identified by genetic screen.** To validate the top 10 most enriched candidate genes (Fig. 2a), the most highly enriched gRNA for each factor was cloned into separate CASAAV vectors. These were used to deplete individually each factor at birth. To focus on cell-autonomous effects

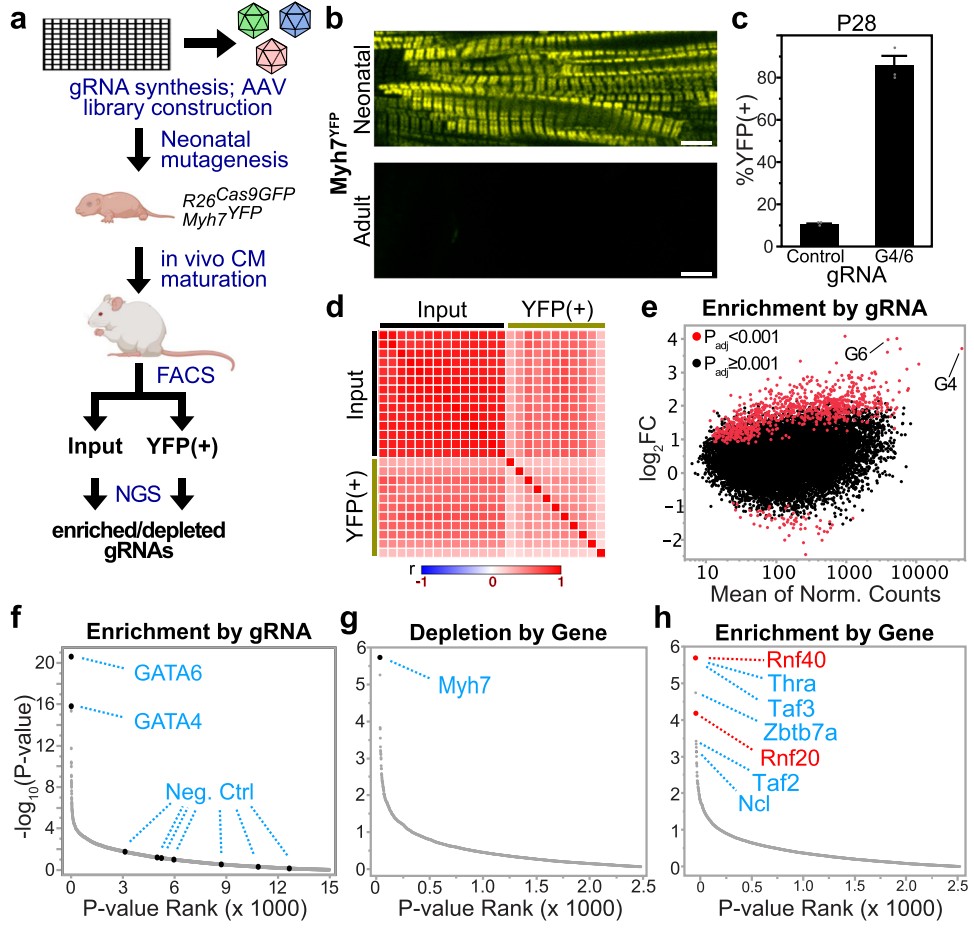

**Fig. 1 High-throughput in vivo CRISPR–Cas9-based forward genetic screen for regulators of cardiomyocyte maturation. a** Schematic overview of the screen design. On-chip oligonucleotide synthesis was used to generate a pooled AAV library containing ~15 k gRNAs that targeted transcriptional regulators. These were delivered to neonatal mice, and a flow cytometry (FACS)-based single-cell screening assay was used to identify immature CMs. Enriched and depleted guides were identified by next-generation sequencing (NGS) of gRNAs. Artwork from biorender.com. **b** Myh7$^{YFP}$ expression is restricted to neonatal stage. Scale bars = 5 μm; $n = 5$ mice. **c** Depletion of both GATA4 and GATA6 by CASAAV resulted in persistent MYH7$^{YFP}$ expression. Data are presented as mean values +/− SEM. $n = 3$ mice per group. **d** Sample clustering by Pearson correlation ($r$). **e** DESeq2 differential expression analysis of individual gRNAs. Red dots are gRNAs with Benjamini–Hochberg corrected Wald test $p$-value < 0.001. GATA4 $P = 1.5 \times 10^{-16}$, GATA6 $P = 2.4 \times 10^{-21}$. **f** Enrichment of positive (GATA4/6) and negative control gRNAs within YFP$^+$ CMs as calculated by DESeq2. **g** Gene depletion as calculated by MAGeCK. **h** Gene enrichment as calculated by MAGeCK.

and avoid secondary effects related to organ dysfunction[5–7], we used an AAV dose that transduced a small fraction of CMs. Among the transduced (GFP +) cells, we observed robust upregulation of Myh7$^{YFP}$ for seven of the ten candidates: *Rnf40*, *Rnf20*, *Taf3*, *Taf2*, *Thra*, *Zbtb7a*, and *Ncl* (Fig. 2a, b; Suppl. Fig. 3a). We focused subsequent analyses on these seven target genes. Of the three candidates that did not validate, one (*Poc1b*) was a false positive as it did not increase YFP expression. The remaining two (*Setd6* and *Eif3i*) caused modest but significant persistence of YFP. Notably, all three of these candidates had less than three gRNAs with DESeq2 enrichment p-value <0.001, suggesting an additional criterion that could improve screen stringency.

We analyzed the effect of individual CASAAV vectors on additional hallmarks of maturation—CM nucleation, size, and T-tubulation. The seven CASAAV vectors, or AAV-Cre lacking gRNA (control), were delivered individually to newborn mice at a mosaic dose. At one month of age, CMs were dissociated, fixed, and stained to visualize T-tubules (CAV3) and nuclei (DAPI). The effects of CASAAV vectors on transduced (GFP +) CMs were compared with AAV-Cre for three additional parameters of CM maturation: multinucleation, maturational hypertrophy, and

T-tubulation. Three of the seven CASAAV vectors, *Rnf40*, *Taf3*, and *Taf2*, negatively impacted all maturation parameters (Fig. 2c–f; Suppl. Fig. 3a–e). CASAAV-Rnf20 impaired maturational growth and T-tubulation, but the increase in mononucleation did not reach statistical significance. *Zbtb7a* impaired T-tubulation, but did not influence maturational hypertrophy or mononucleation. These results validated the requirement of seven of the 10 candidates for multiple facets of CM maturation, and also demonstrate that the overall maturational program can be separated into independently regulated, dissociable subprograms[7].

**RNF20/40 are required for regulation of CM maturation.** *Rnf20* and *Rnf40* were two of the most enriched genes in the screen, and their depletion broadly impaired CM maturation. These genes encode subunits of an E3 ubiquitin ligase that monoubiquitinates histone 2B at lysine 120 (H2Bub1)[18,29], an epigenetic mark that regulates the expression of a subset of genes and that has been implicated in developmental transitions and cancer[30]. Human genetic studies implicate de novo *RNF20/RNF40* mutations in congenital heart disease[16,17]. The postnatal cardiac functions of these genes have not been studied. For these reasons, we

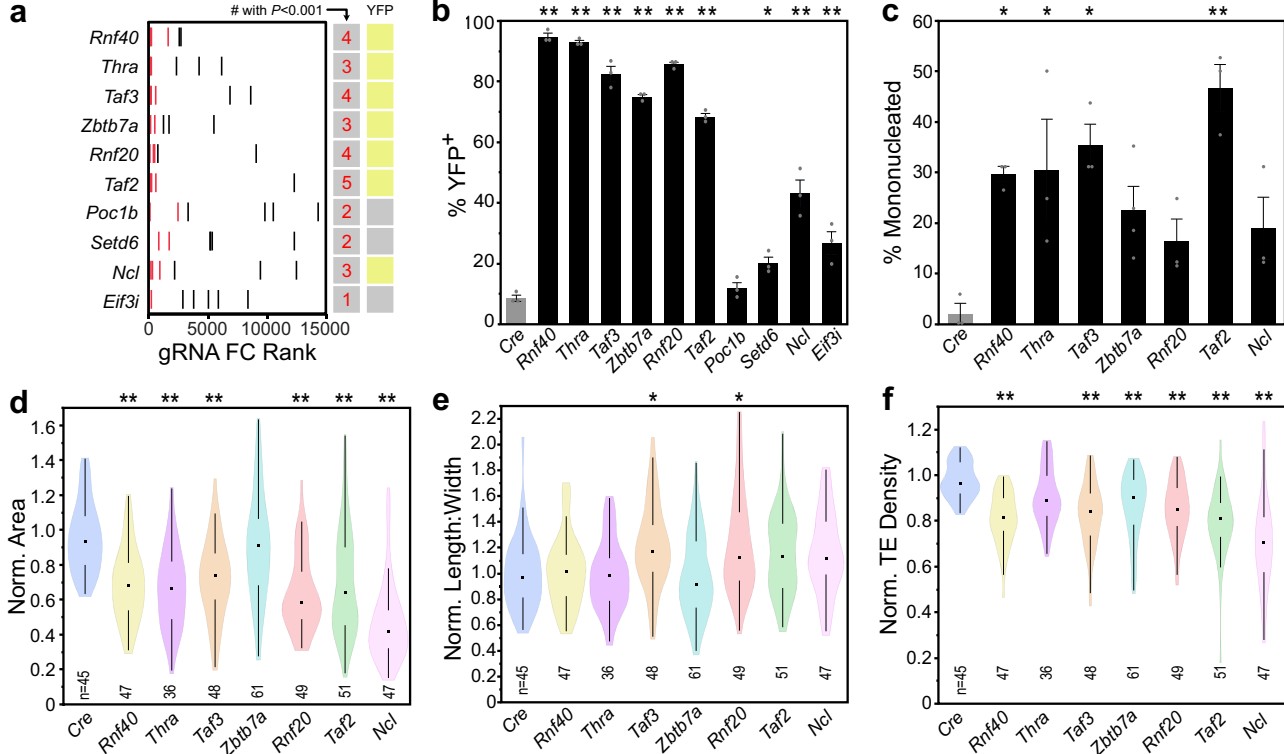

**Fig. 2 Validation of top candidates from the forward genetic screen. a** Top ten candidates ranked by MAGeCK enrichment score (lower rank indicates greater enrichment). The six guides for each candidate are ranked by individual enrichment. Enriched gRNAs with Benjamini–Hochberg corrected Wald test $P < 0.001$ are colored in red and summed in the first column. Candidates were tested by treating $R26^{Cas9-GFP}$; $Myh7^{YFP}$ neonatal pups with the most highly ranked individual gRNA. AAV-Cre without gRNA served as negative control. Presence (column 2, yellow box) or absence (gray box) of persistent YFP expression at four weeks was assessed by imaging fixed, dissociated CMs. **b** Quantification of $Myh7^{YFP}$ activation within transduced (GFP + ) CMs by flow cytometry. AAV-Cre was used as negative control. $n = 3$. $P$ for all genes, except Poc1b and Setd6, was <0.0001. Setd6 $P = 0.0111$. **c** Quantification of mononucleation among dissociated, GFP + control CMs or YFP + candidate-depleted CMs. $n = 3$. Significant $P$: Rnf40, 0.0128; Thra, 0.0102; Taf3, 0.0027; Taf2, 0.0001. Error bars denote standard error. **d–f** Normalized projected area (**d**) and length:width ratio (**e**) of control and candidate-depleted CMs. Measurements from dissociated YFP + CMs were normalized to YFP- cells from the same heart. Significant $P$, area: Rnf40, Thra, Taf3, Rnf20, Taf2, and Ncl <0.0001. Significant $P$, length-to-width: Taf3, 0.007; Rnf20, 0.0042. **f** T-tubule transverse element density for control and candidate-depleted CMs as measured by AutoTT software-based quantification of CAV3 immunostaining. Significant $P$: Rnf40, Taf3, Rnf20, Taf2, Ncl < 0.0001; Zbtb7a, 0.0003. Bar plots show mean ± SD. At least 69 CMs from three mice were measured per group. Violin plots: shape indicates data distribution; point, median; whiskers, starts at quartile and extends 1.5 times the interquartile distance. Dunnett's two-tailed $t$-test vs. Cre control: *$P < 0.05$, **$P < 0.001$. Source data are provided as a Source Data file.

investigated the mechanisms by which RNF20/40 regulate CM maturation.

We investigated the expression profile and transcriptional regulation of *Rnf20/40*. Mining of a single-cell RNA-seq data set from the adult mouse heart revealed that Rnf20 and Rnf40 are expressed in CMs and several other cell populations in the adult mouse heart (Suppl. Fig. 4a–c)[31,32]. In cardiomyocytes, *Rnf20* and *Rnf40* transcript levels were similar between neonatal and adult stages (Suppl. Fig. 4d). Examination of our previously reported core cardiac transcription factor ChIP-seq datasets[33] demonstrated strong binding of multiple transcription factors, including GATA4, NKX2–5, and SRF, at the promoters of both genes (Suppl. Fig. 4e, f). Consistent with our observation of stable postnatal gene expression, we did not observe significant differences in transcription factor binding at either locus between fetal and adult stages. These results could suggest developmental post-transcriptional regulation of RNF20/40 function, developmental regulation of cofactors, or developmental regulation of RNF20/40 recruitment to regulate genes.

We created a single CASAAV vector (CASAAV-RNF20/40) containing gRNAs that target both factors and validated that it depleted RNF20/40 (RNF Cas-KO) and H2Bub1 (Fig. 3a, b).

H2Bub1 was more depleted than RNF20/40, possibly because some CRISPR mutations in *Rnf20* or *Rnf40* might impair activity but not immunoreactivity, and because RNF20 and RNF40 are likely both required for enzymatic activity to write H2Bub1[18]. When given at a dose that transduced most CMs ("full dose"), CASAAV-RNF20/40 caused cardiac dysfunction, dramatic elevation of $Myh7^{YFP}$ expression, and death (Fig. 3c–e). At a dose that transduced a small fraction of CMs ("mosaic dose"), RNF20/40 depletion cell autonomously impaired CM maturation (Fig. 3f–n).

As CMs mature, they become polyploid, making simultaneous mutagenesis of all alleles by CASAAV-based indel formation more difficult. To bypass this difficulty, and further validate our CASAAV-based findings, we acquired a conditional *Rnf20* allele, $Rnf20^{fx}$[34]. Injection of $Rnf20^{fx/fx}$; $R26^{fsTomato}$ newborn pups with a mosaic dose of AAV-TnT-Cre resulted in robust depletion of H2Bub1 within transduced CMs (*Rnf20* KO; Suppl. Fig. 5a, b). Subsequent analysis of CM area, length, width, and T-tubule organization revealed an immature phenotype consistent with our CASAAV-based observations (Suppl. Fig. 5c–i). To determine if the dependence of mature CM phenotype on RNF20 is developmental stage specific, we treated $Rnf20^{flox}$; $R26^{fsTomato}$

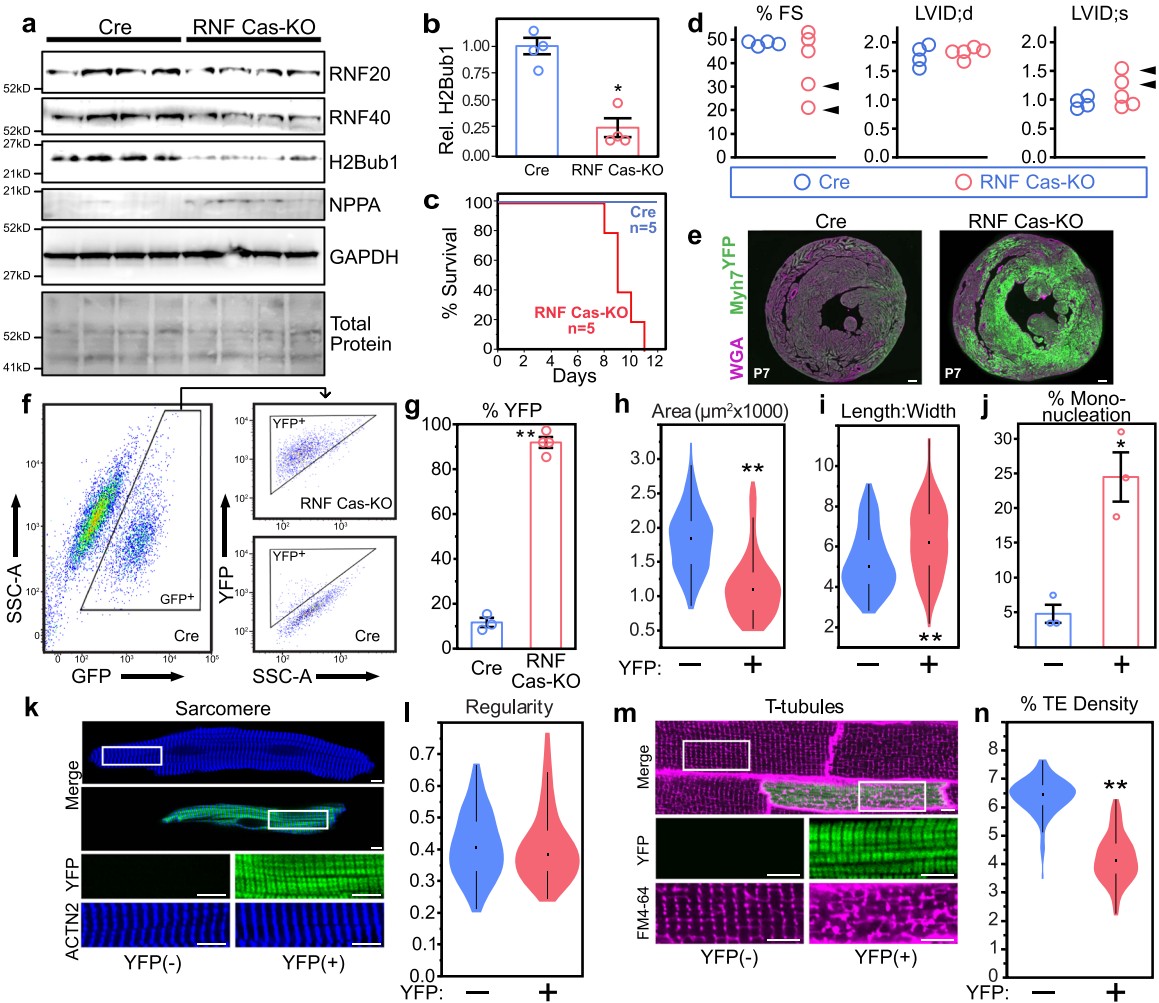

**Fig. 3 Characterization of CASAAV-RNF20/40 and RNF20/40-depleted CMs. a–d** Neonatal pups were treated with CASAAV-RNF20/40 (RNF Cas-KO) at a dose that transduced most CMs. Western blotting of ventricular apexes (**a**) was performed to measure protein levels of RNF20, RNF40, H2bub1, and NPPA, with GAPDH internal control. RNF20, RNF40, and H2bub1 were depleted, whereas NPPA was upregulated. n = 4 mice per group. Quantification of H2bub1 (**b**) showed significant 74% reduction (P = 0.0007), consistent with reduced RNF20/40 ubiquitin ligase activity. Reduction in tissue samples with multiple cell types underestimates changes in cardiomyocytes. Survival curve (**c**) demonstrated death of juvenile RNF Cas-KO mice in the majority of CMs. At P7, before the onset of lethality, echocardiography (**d**) showed that a subset of mice (arrowheads) exhibited cardiac dysfunction (reduced fractional shortening percentage and increased LV internal diameter at end systole). **e** Sections of RNF Cas-KO hearts (n = 3 mice) at P7 showed dramatic upregulation of Myh7$^{YFP}$ compared with controls (n = 3 mice). Scale bars = 200 μm. **f–n** In order to avoid non-cell-autonomous secondary effects of heart failure, CASAAV-RNF20/40 was administered to newborn R26$^{Cas9-GFP/+}$;Myh7$^{YFP/+}$ pups at a mosaic dose. Analysis was performed at P28. **f** Representative flow cytometry analysis. Cells were gated on GFP (transduction marker) and then on YFP. **g** Quantification of YFP + transduced CMs. RNF Cas-KO markedly increased the fraction of transduced CMs that retained Myh7$^{YFP}$ expression at P28 (92% of CASAAV-RNF20/40-transduced CMs, compared with 13.5% of AAV-Cre; P < 0.0001). **h–n** Analysis of CM maturation. RNF Cas-KO CMs were markedly smaller than controls (**h**; P < 0.0001), with a moderate increase in length-to-width ratio (**i**; P = 0.0002). **j** These RNF20/40-depleted CMs had increased mononucleation (P = 0.0207). **k–l** However, sarcomere organization appeared unperturbed. In total 89 YFP- and 93 YFP + CMs were randomly selected from three mice in equal proportions for quantification. **m**, **n** T-tubules were markedly disrupted, as determined by optical sectioning of freshly isolated, FM4-64-perfused hearts followed by quantitative analysis of T-tubule transverse element (TE) density (P < 0.0001). In total 79 YFP- and 69 YFP + CMs were randomly selected from three mice in equal proportions for quantification. Collectively, these phenotypes observed in RNF Cas-KO CMs indicate that maturation is impaired. Source data are provided as a Source Data file. Scale bars in k and m = 5 μm. Violin plots: shape indicates data distribution; point, median; whiskers, starts at quartile and extends 1.5 times the interquartile distance. Bar plots: error bars denote standard error. Two-tailed t-test: *P < 0.05, **P < 0.001.

mice with a mosaic dose of AAV-TnT-Cre at eight weeks of age and dissociated CMs for analysis at 12 weeks. Area, length, and width of CMs were unaffected by mosaic RNF20 KO (Suppl. Fig. 5j–l). In contrast, T-tubule organization was disrupted, with KO CMs demonstrating decreased transverse element density and increased longitudinal element density (Suppl. Fig. 5m–p). These data indicate that RNF20 is necessary in adult CMs for maintenance of some features of maturation (T-tubules) but not others (size and overall morphology).

**Transcriptional regulation of CM maturation by RNF20/40.** To measure the effect of RNF20/40 depletion on gene expression during maturation, we conducted transcriptional profiling. Newborn R26$^{fsCas9-2A-GFP/+}$;Myh7$^{YFP/+}$ mouse pups were injected with a mosaic dose of CASAAV-RNF20/40 vector or a control vector containing Cre without gRNAs. At P28, hearts were dissociated and flow cytometry was used to recover YFP$^+$ CMs transduced with CASAAV-RNF20/40 or control GFP$^+$ CMs transduced with Cre. We analyzed the transcriptomes of each

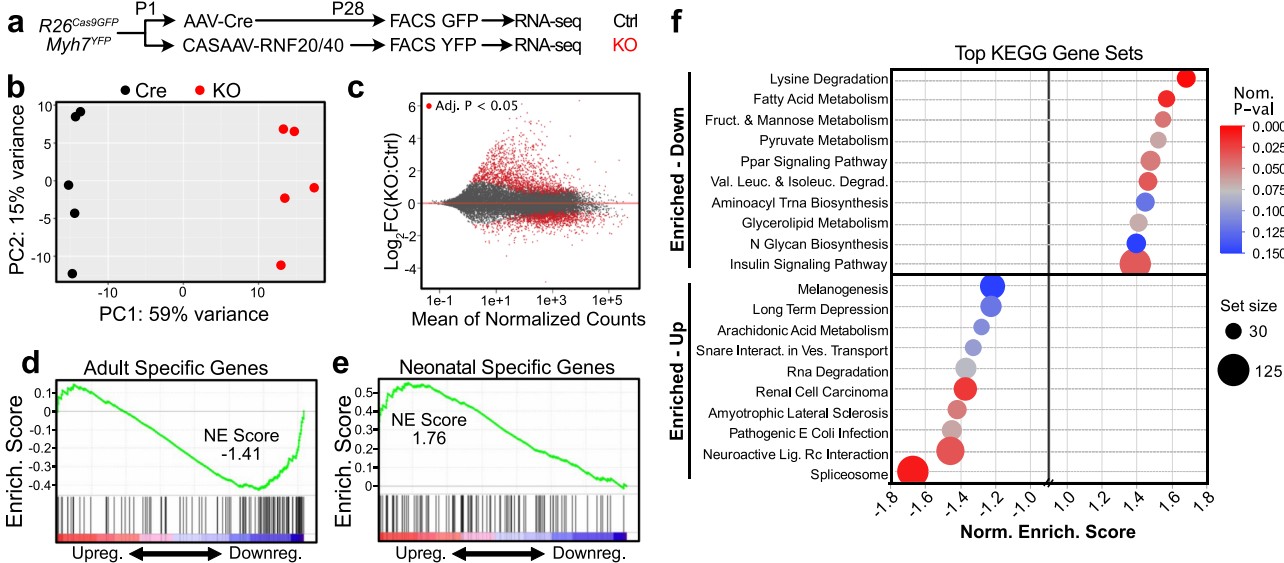

**Fig. 4 Transcriptional profiling of mosaic *RNF20/40*-depleted CMs.** a Experiment overview. **b** Principal component analysis of RNA-seq replicates. **c** Differentially expressed genes. Red indicates Benjamini–Hochberg-corrected Wald test $P < 0.05$. **d, e** Gene set enrichment analysis with custom gene sets for "Adult Specific Genes" (**d**) or "Neonatal Specific Genes" (**e**). RNF20/40-depleted cells were depleted for adult-specific genes and enriched for adult-specific genes. **f** Gene set enrichment analysis of indicated KEGG pathway gene sets, using genes ranked by their expression in RNF20/40-depleted CMs compared with control CMs. GSEA nominal P-val is not corrected for set size or multiple-hypothesis testing, the Normalized Enrichment Score. Source data are provided as a Source Data file.

group by RNA-seq (Fig. 4a). Principal component analysis showed clear separation of sample groups (Fig. 4b), and differential gene expression analysis revealed approximately 1400 upregulated and 1100 downregulated genes ($P_{adj} < 0.05$, Fig. 4c, and Suppl. Data 4). As expected, *Rnf20* and *Rnf40* were among the downregulated genes ($P_{adj} = 0.029$ and $<0.001$, respectively). The ratio of fetal (*Tnni1*) to mature (*Tnni3*) troponin I isoforms, a molecular signature of CM maturational state[35], was increased by 165-fold in RNF Cas-KO CMs ($P < 0.001$), with *Tnni1* being the most upregulated gene in the dataset. To assess the genome-wide impact of RNF20/40 depletion on the maturational status of the CM transcriptome, we assessed the enrichment of RNF20/40 differentially expressed genes among the 100 most neonatal- or adult-biased genes (Suppl. Fig. 6, Suppl. Table 1, Suppl. Data 5) using Gene Set Enrichment Analysis (GSEA)[36]. Genes downregulated in RNF20/40-depleted CMs were highly enriched for adult-biased genes (normalized enrichment score (NES) = −1.41; Fig. 4d), while upregulated genes were highly enriched for neonatal-biased genes (NES = 1.76; Fig. 4e). These data confirm that RNF Cas-KO CMs fail to activate the transcriptional network of mature CMs and instead persistently express genes associated with immaturity.

We next analyzed the RNA-seq data to identify biological processes enriched within the RNF Cas-KO differentially expressed genes (Suppl. Table 2). Many metabolism-related gene sets were enriched within the downregulated genes (Fig. 4f), including fatty acid metabolism and PPAR signaling, two pathways strongly associated with CM maturation[37,38]. Upregulated genes were enriched for a greater diversity of biological processes, with spliceosome-associated genes being most enriched (Fig. 4f).

### RNF20/40 regulation of the maturational epigenetic landscape.
The RNF20/40 complex is a E3 ubiquitin ligase that mono-ubiquitinates H2B on lysine 120 (H2Bub1). To determine the relationship between H2Bub1 written by RNF20/40 and normal maturation of CM gene expression, we used chromatin immunoprecipitation followed by next-generation sequencing (ChIP-seq) to measure H2Bub1 chromatin occupancy at neonatal (first week of life) and mature (four-week-old) stages in mouse heart apex tissue. Biological duplicates at each stage correlated well with one another (Fig. 5a and Suppl. Fig. 7a). Consistent with prior reports[39,40], H2Bub1 predominantly occupied gene bodies, with greater density toward promoters (Fig. 5a, b, Suppl. Fig. 7b). We quantified H2Bub1 density on gene bodies (Suppl. Data 6). The vast majority (>88%) of genes with detectable H2Bub1 signal were shared between time points (Fig. 5c), although the signal intensity of a dynamic subset varied between developmental stages (Fig. 5a, d). Consistent with H2Bub1 being an activating mark[41–43], at both stages, genes with greater H2Bub1 generally were more highly expressed, although the correlation was poor ($r = 0.07–0.09$; Suppl. Fig. 7c, d). Interestingly, genes with stronger H2Bub1 signal at P1 (or P28) were less likely to be differentially expressed between these stages (Fig. 5e, top plot; Suppl. Fig. 7e), consistent with observations in murine embryonic fibroblasts[44]. Gene sets enriched by greater H2Bub1 occupancy included oxidative phosphorylation and TCA cycle (Suppl. Fig. 7f–g).

Although gene expression and H2Bub1 occupancy at each stage were not well correlated, there was a significant correlation between the change in H2Bub1 and the change in gene expression between stages ($r = 0.39$; $P < 0.0001$; Fig. 5e, bottom plot). Genes upregulated in RNF Cas-KO (red, Fig. 5f and Suppl. Fig. 7e) overall were more highly expressed in immature cardiomyocytes, and conversely those downregulated in RNF Cas-KO (blue, Fig. 5f and Suppl. Fig. 7e) were overall more highly expressed in mature cardiomyocytes. Among genes that were up- or downregulated by more than 2-fold in RNF20/40 depletion, 295 genes gained H2Bub1 and increased expression during maturation (upper right quadrant, Fig. 5f). Of these, 218 were downregulated by inactivation of RNF20/40 (blue color versus 398 of 985 genes in the other three quadrants: Fisher exact test $P < 0.00001$). Among these 218 genes were *Myh6* and *Tnni3*, whose well-established isoform switches are cardinal features of maturation[35]. These 218 genes were highly enriched for functional terms related to metabolism (Fig. 5g, right). Conversely, genes that lost H2Bub1

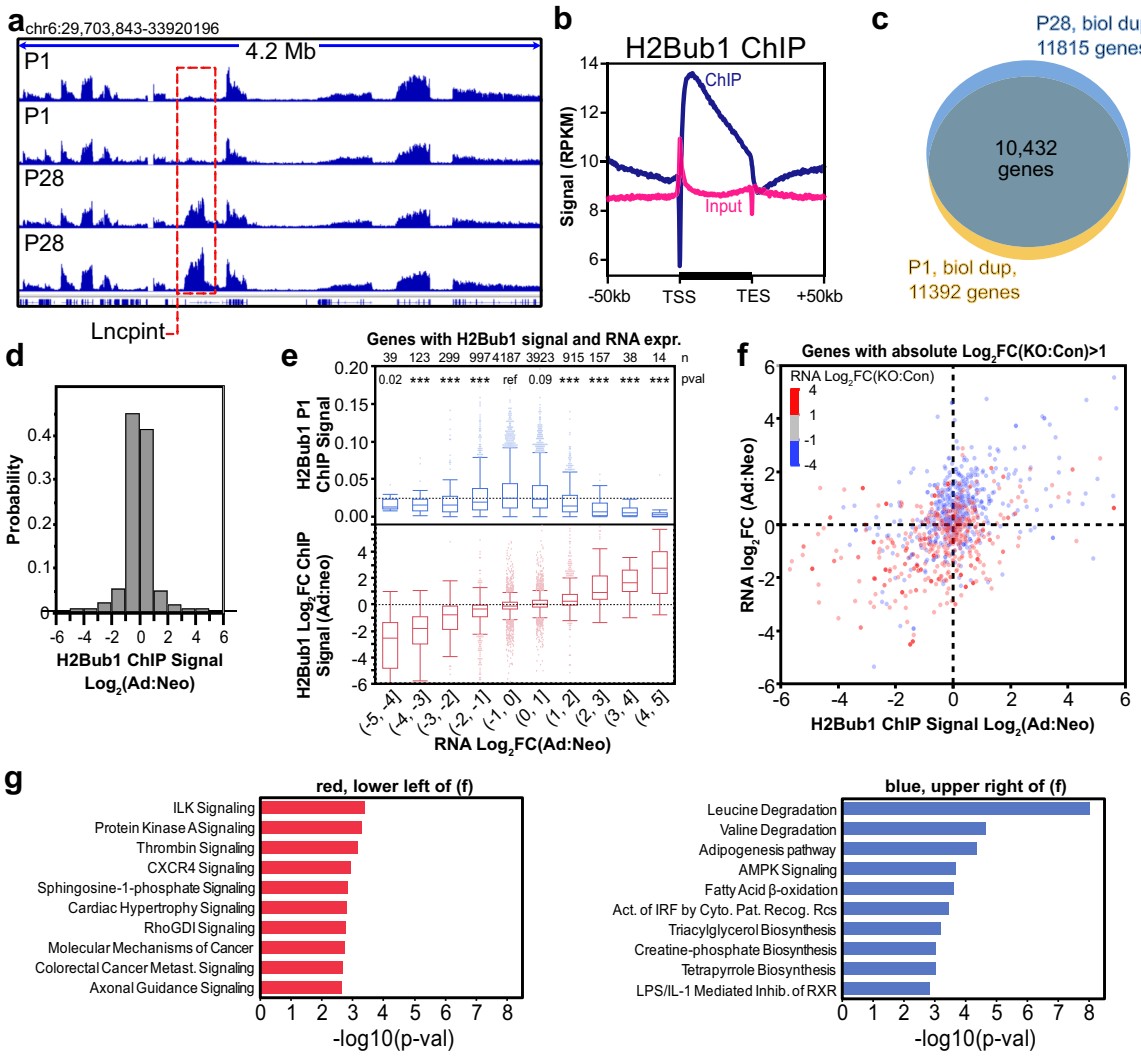

**Fig. 5 H2Bub1 regulation of maturational gene expression. a** Representative genome browser view of neonatal and adult H2Bub1 ChIP-seq replicates. Dashed red box highlights a gene with dynamic H2Bub1 occupancy. **b** Profile plot showing average H2Bub1 signal at gene bodies. Plot shows gene bodies and 50 kb upstream of the TSS or downstream of the TSE. **c** Genes with detectable H2Bub1 ChIP-seq signal at P1, P28, or both timepoints. **d** Distribution of changes in H2Bub1 signal intensity during maturation, for all genes marked by H2Bub1 in at least one timepoint. **e** H2Bub1 P1 ChIP signal (top) or the change of H2Bub1 ChIP signal during maturation (bottom) as a function of the gene's maturational change in RNA expression. The change in RNA was binned to contain the indicated range of log2 fold-change values. The change in gene expression correlated with the change in H2Bub1 occupancy, but not with the absolute H2Bub1 occupancy. Numbers at the top of plot (n) indicate genes per bin. Center lines in box plots indicate the median, while boxes show 25th and 75th percentiles. Whiskers denote the maximum observation within the 75th percentile +1.5 times the interquartile range, or the minimum observation within the 25th percentile −1.5 times the interquartile range. P-value: Steel–Dwass vs. H2Bub1 P1 ChIP signal in the (−1,0] bin; ***$P < 0.001$. **f** Maturational change in expression versus maturational change in H2Bub1 signal plotted for genes differentially expressed in RNF20/40-depleted CMs. Color indicates the direction of differential expression in RNF20/40-depleted CMs. Genes with maturational gain in H2Bub1 and RNA expression were predominantly downregulated in RNF20/40 depletion (blue, upper-right quadrant), whereas genes with maturational loss in H2Bub1 and RNA expression were mostly upregulated in RNF20/40 depeletion (red, lower-left quadrant). **g** Ingenuity pathway analysis of functional terms enriched amongst the indicated genes. Left: Red-colored genes in the lower-left quadrant of panel **f** (i.e., genes that decrease H2Bub1 and gene expression during maturation and are upregulated in RNF20/40 depletion). Right: Blue-colored genes in the upper-right quadrant of panel **f** (i.e., genes that increase H2Bub1 and gene expression during maturation and are downregulated in RNF20/40 depletion). P-values were calculated by IPA Core Analysis. Source data are provided as a Source Data file.

and decreased expression during maturation were highly enriched for upregulation by RNF20/40 inactivation (167 of 261 in lower left quadrant of Fig. 5f, compared with 497 of 1019 in the other three quadrants; Fisher exact test $P < 0.00001$). Gene ontology analysis of these 167 genes showed that they were enriched for diverse cell-signaling functional terms, such as protein kinase A signaling and cardiac hypertrophy signaling (Fig. 5g, left).

H3K4me3 and H3K36me3 are epigenetic marks that correlate with H2Bub1[44,45]. Although H2Bub1 was required for H3K4me3

modification in yeast[46,47], this effect was less pronounced and confined to specific genes in mammalian cells[45]. To study the effect of RNF20 inactivation on H3K4me3 and H3K36me3 chromatin occupancy in maturing CMs, we treated $Rnf20^{fx/fx}$ neonates with a full dose of AAV-TnT-Cre to inactivate $Rnf20$ in the large majority of CMs ($Rnf20$ KO). Wild-type pups injected with a high dose of AAV-TnT-Cre were used as controls (WT). H3K4me3 and H3K36me3 ChIP-seq was performed on heart apexes at P7. As expected, H3K4me3 and H3K36me3 consistently

 

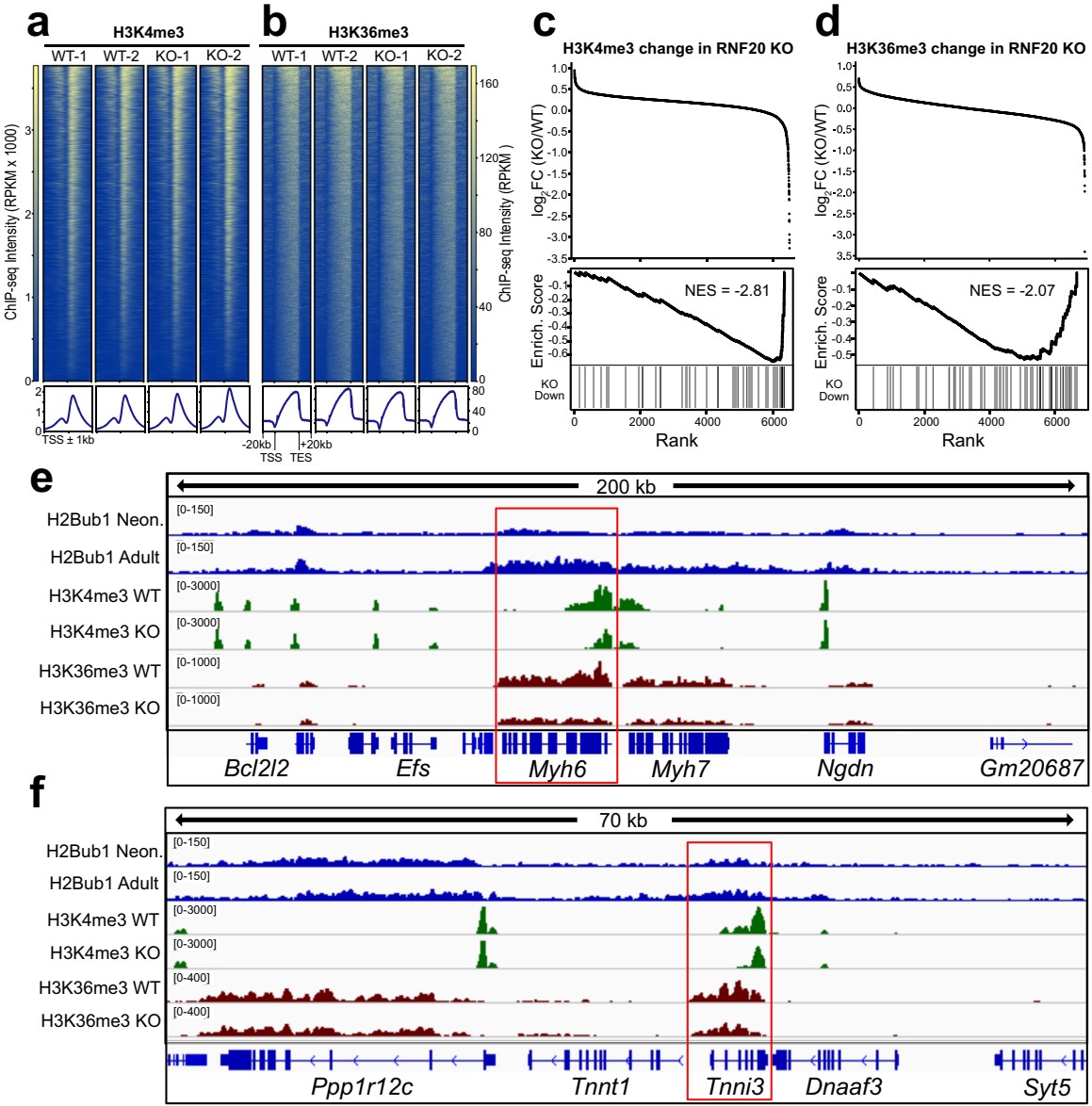

**Fig. 6 Impact of *Rnf20* inactivation on H3K4me3 and H3K36me3.** Wild-type or *Rnf20fx/fx* newborn pups were injected with a high dose of AAV9-TnT-Cre. P7 heart apexes were used for H3K4me3 and H3K36me3 ChIP-seq. **a, b** H3K4me3 and H3K36me3 marked active TSSs and gene bodies in control and knockout hearts. The overall distribution pattern of these marks was not altered by *Rnf20* inactivation. **c, d** H3K4me3 signal at each TSS and H3K36me3 signal at each gene was quantified in wild-type and *Rnf20* KO groups (upper panel). GSEA revealed that the set of genes that were downregulated in RNF Cas-KO were strongly enriched for genes with reduced H3K4me3 or H3K36me3 signal in *Rnf20* KO (lower panel; *P* < 0.001; NES = normalized enrichment score calculated by GSEA that corrects for gene set size). **e, f** IGV genome browser views at *Myh6* and *Tnni3* (red boxes), which are involved in maturational sarcomere isoform switching. These genes were downregulated in RNF Cas-KO and had reduced H3K4me3 and H3K36me3 occupancy in *Rnf20* KO. Source data are provided as a Source Data file.

marked promoters and gene bodies, respectively, of actively transcribed genes in all replicates (Fig. 6a, b). H3K4me3 and H3K36me3 signal at each TSS and gene body, respectively, was quantified in WT and *Rnf20* KO (Fig. 6c, d upper panels). Genes with reduced H3K4me3 or H3K36me3 signal in *Rnf20* KO were strongly enriched for the set of genes downregulated in RNF Cas-KO (Fig. 6c, d lower panels). Among these were the mature sarcomere isoform genes *Myh6* and *Tnni3*, which were downregulated in RNF Cas-KO and had reduced H3K4me3 and H3K36me3 occupancy in *Rnf20* KO (Fig. 6e, f). Additional gene sets, such as the maturation genes identified in Fig. 5f–g, were also significantly enriched among genes with reduced H3K4me3 and H3K36me3 occupancy (Suppl. Fig. 8a, b; gene set "Intersect. mature"). The correlation between change in gene expression in

RNF Cas-KO and change in H3K4me3 or H3K36me3 occupancy was poor (Suppl. Fig. 8c, d), suggesting that changes in these epigenetic marks are not simply the downstream effects of altered gene expression. Together, these analyses identify RNF20/40 as the first epigenetic regulators of CM maturation.

## Discussion
In this study, we developed a resource-efficient, in vivo, forward genetic screen for transcriptional regulators of CM maturation. Our strategy uses a massively parallel approach in which the screening unit is the individual cell, such that thousands of individual genetic mutations can be screened using a small number of animals. This strategy will be broadly applicable for

cells efficiently transduced by AAV and for phenotypes amenable to high-throughput, single-cell measurement. For example, the forward genetic screen described here could be readily extended to study pathological cardiomyocyte hypertrophy using the same Myh7-YFP reporter, which is reactivated by pathological cardiomyocyte stress. The scope of accessible phenotypes could be dramatically expanded by developing methods to sort individual cells by morphology[48].

Among the validated transcriptional and epigenetic regulators of CM maturation uncovered by our screen are *Rnf20*, *Rnf40*, *Taf2*, and *Taf3*. Notably, these genes are components of protein complexes whose perturbation is linked to congenital heart disease. Rnf20 and Rnf40 encode subunits of a H2B ubiquitin ligase, and CHD patients were enriched for de novo mutations in genes in this H2B monoubiquitination pathway, including RNF20 and UBE2B, an E2 ubiquitin-conjugating enzyme involved in writing H2Bub1[16,17]. *Taf2* and *Taf3* are TATA-binding protein-associated factors, components of the TFIID protein complex that positions RNA Polymerase II at gene promoters. Another member of this complex, *Taf1*, was found to have a significant burden of predicted damaging mutations in CHD probands compared with controls[49]. This overlap between genes that regulate CM maturation and genes that cause CHD likely reflects the important function of these genes in both heart morphogenesis and cardiomyocyte maturation. One implication of this overlap is that patients with CHD caused by mutations in these genes may be at risk for later complications, such as myocardial dysfunction, due to impaired CM maturation.

*Rnf20* and *Rnf40* interact to form a ubiquitin ligase complex that writes the H2Bub1 epigenetic mark. Depletion of *Rnf20/40* broadly impaired CM maturation by abrogating writing of H2Bub1 and preventing essential transcriptional changes. H2Bub1 has been implicated in regulating cell-state transitions during development and cancer. Consistent with our results, a previous study in murine embryonic fibroblasts found that genes differentially expressed following RNF40 depletion were those with low or moderate H2Bub1[44]. Strongly activated genes with high H2Bub1 may not be dependent on H2Bub1 because of redundant activating mechanisms, whereas a subset of genes with lower H2Bub1 may be more reliant upon H2Bub1-dependent mechanisms[44]. Further experiments are needed to understand what makes the expression of specific genes, such as genes activated during cardiomyocyte maturation and other developmental transitions, dependent upon RNF20/40 and H2Bub1. In addition, nonhistone RNF20/40 substrates have been identified[50–53], and it is possible that they also contribute to gene regulation of CM maturation.

## Methods

**Mice**. All relevant ethical regulations for animal testing and research were followed. All procedures were performed following protocols approved by the Boston Children's Hospital Institutional Animal Care and Use Committee. Mice were maintained at 65–75°F with 40–60% humidity on a normal 12-hour light, 12-hour dark cycle. *Myh7*[YFP], *R26*[fsCas9-P2A-GFP], *Rnf20*[flox], and *R26*[fsTomato] mice were described previously[13,20,34,54]. Echocardiography was performed on a VisualSonics Vevo 2100 machine with the Vevostrain software. Animals were awake during this procedure and held in a standard handgrip. The echocardiographer was blinded to genotype and treatment.

**Library design and construction**. A list of 3000+ potential transcription factors and epigenetic regulators was compiled using publicly available data from the Riken Transcription Factor Database[55], and AnimalTFDB[56]. RNAseq data from isolated P6 wild-type CMs were used to prioritize 1894 factors. In addition, 259 genes differentially expressed at P6 within GATA4/6 double-KO CMs (adjusted *p*-value < 0.05), and 291 genes with strong adjacent GATA4 binding at P0 were also included, for a total of 2444 genes targeted. Six gRNAs were selected for each gene using CRISPR RGEN Tools Cas-Database[23]. To select the "best" six guides per gene, the following five rules were applied, in order, by sequentially applying each rule if the candidate set contained less than six guides: (1) gRNAs target

constitutive exons, CDS range 5–50%, number of mismatches 0,1,2:1,0,0 (i.e., only one on-target exact match, zero off-target sites with one mismatch, and zero off-target sites with two mismatches, and microhomology-associated out-of-frame score >60); (2) gRNAs target constitutive exons, CDS range 5–80%, number of mismatches 0,1,2:1,N,N and out-of-frame score >60; (3) gRNAs target any exon, CDS range 5–80%, number of mismatches 0,1,2:1,N,N and out-of-frame score >60; (4) gRNAs target any exon, CDS range 5–80%, number of mismatches 0,1,2:N,N,N and out-of-frame score >60; (5) gRNAs target any exon, any CDS range, number of mismatches 0,1,2:N,N,N, and any out-of-frame score. A 5′ G was added to each gRNA to ensure optimal expression from the U6 promoter, and gRNAs were flanked by SapI restriction sites. Seven human gRNAs that are not predicted to target the mouse genome were also included. The 14,675 guide library was synthesized by Agilent as a single 80-nt SUREprint oligo pool (Suppl. Data 2). The library was resuspended in 50 μl of TE (200 nM) and diluted to 33 nM. About 1 μl of 33 nM library was amplified for 10 cycles using a standard NEB Phusion PCR program and 80-nt library amplification primers (Suppl. Table 3), to produce a 200-bp amplicon. Eight reactions were pooled, cleaned up via the DNA Clean and Concentrator Kit (Zymo, D4014), and 5 μg were digested with SapI for 3 h. The 21-bp gRNA library was purified via Invitrogen Size Select 2% gel, and seamlessly ligated into our previously described CASAAV vector (Addgene 132551)[5,57]. The ligation product was purified via Zymo DNA column and electroporated into Agilent SURE Electrocompetent cells with a Bio-Rad Gene Pulser Xcell Electroporation System (1700V, 200-ohms resistance, 25-μF capacitance, and 1-mm cuvette gap). In all, 40 ng of purified ligation product in 2-μl volume was electroporated into 40 μl cells. About 900 μl of SOC media was added immediately after electroporation, and bacteria were incubated at 37 °C for 1 hr. Bacteria were plated on LB agar containing ampicillin and allowed to grow for 18 h. Four electroporations collectively yielded ~300,000 colonies (20x library coverage). Colonies were scraped into SOC media, cultured for an additional 1.5 h, and plasmid DNA harvested (Invitrogen Purelink HiPure Maxiprep, 210017), yielding 240 μg of DNA. The gRNA plasmid library has been deposited at Addgene (Cat.# 138015). This library pool was packaged into AAV9 via PEI transfection of ten 15-cm plates of HEK293T cells (Agilent, 240073). AAV was titered by qPCR targeting the TnT promoter, using standard qPCR parameters[5,57].

**Injections, sample collection, and flow cytometry**. One-day-old *R26*[fsCas9-2A-GFP/+]; *Myh7*[YFP/+] pups were subcutaneously injected with 50 μl of gRNA library virus at a concentration of $2 \times 10^{11}$ vg (viral genomes) per ml and spiked with a single CASAAV virus targeting both GATA4 and GATA6 at a final concentration of $1 \times 10^9$ vg/ml. At four weeks of age, hearts were digested by retrograde perfusion of 2 mg/ml Collagenase-II (Worthington LS004177) in 37 °C perfusion buffer (120 mM NaCl, 14.7 mM KCl, 0.6 mM KH$_2$PO$_4$, 1.2 mM MgSO$_4$–7H$_2$O, 10 mM Hepes, 4.6 mM NaHCO$_3$, 30 mM Taurine, 10 mM BDM, and 5.5 mM Glucose) for 12 min[5,57]. Atria were removed and discarded after perfusion, and the ventricles were dissociated to single CMs. About 15% of the isolated CMs (~200,000) were set aside as an unsorted input sample, while YFP$^+$ CMs were sorted from the remaining 85% via FACS. Sorting was performed at the DANA Farber Cancer Center flow cytometry core on an Aria II cell sorter with 100-μm nozzle, 510/21 band-pass filter for GFP, 550/30 band-pass filter for YFP, and 525 long-pass dichroic filter to split GFP and YFP signals. CMs were sorted into Trizol (Life Technologies, 15596026). In total, approximately 365,000 YFP + CMs were collected from 4 M transduced CMs. The plots in Supplemental Fig. 2a are saved events directly from screen and control samples. The GATA4/6 double-KO positive control and Cre-only negative control were used to set the YFP gating of GFP + CMs. An example of typical GFP gating is shown in Supplemental Fig. 4E. RNA was extracted from sorted YFP + and input CMs via Trizol. CMs from three hearts were sorted into each RNA collection tube (45 hearts, 15 pooled samples). Trizol aqueous phase was transferred to a Zymo RNA Clean and Concentrator spin column (Zymo, R1015) and treated with DNase (Qiagen, 79254) for 20 min, followed by cleanup, and elution in 15 μl of H$_2$O for YFP samples and 120 μl of H$_2$O for input samples. After isolation, input RNA samples were pooled in groups of three to match YFP samples.

**NGS library preparation**. Reverse transcription of gRNAs and adapter addition was achieved using a custom protocol with the Clontech SMART-Seq v4 Ultra Low Input RNA sequencing kit (634894). Approximately 30 ng of RNA from YFP$^+$ CMs or 500 ng RNA from input CMs was reverse transcribed as directed in the manufacturer's protocol, except that SMART-Seq CDS Primer IIA was replaced with a gRNA scaffold-specific primer (Suppl. Table 3). This reverse-transcription step utilizes a template-switching oligo to add an adapter of known sequence to the variable 5′ end of the gRNA[58]. cDNA was then amplified in two sequential rounds of PCR to add NGS sequencing adapters. In the first round of amplification, the full-length forward-read adapter was added to the 5′ end of the gRNA and a half-adapter was added to the 3′ end (Suppl. Table 3). NEB Phusion (M0530L) was used for five cycles of amplification, according to the manufacturer's protocol. First-round PCR product was purified via Zymo DNA Clean and Concentrator column and eluted in 25 μl of H$_2$O. About 5 μl of the purified product was used as input in a second round of 20 cycles of amplification, which completed the reverse adapter and added a sample-specific Illumina TruSeq multiplexing index. The resulting 220-bp amplicon was purified via Invitrogen SizeSelect 2% gel. The concentration of each sample was assessed using the KAPA Library Quantitation Kit (KR0405), allowing samples to be evenly pooled and submitted for single-end 75-nt

sequencing on a NextSeq500. An average sequencing depth of 4.8 million reads per sample was achieved.

**Screen NGS analysis**. gRNA NGS libraries were trimmed to remove adapters and the gRNA scaffold, leaving only the 20-bp variable region. Bowtie2 was used to align trimmed sequence files to mm10. Counts for each gRNA were acquired by quantifying sequence coverage of genomic regions corresponding to the gRNA library via Bedtools Coverage. Differential expression of individual gRNAs in YFP+ versus Input samples was calculated by using gRNA counts as input for DESeq2[59]. Differential gene representation in YFP+ versus input samples was calculated using median-normalized gRNA counts as input for the MAGeCK software package[24], which consolidates scores for multiple individual gRNAs targeting the same gene into a single-gene-level enrichment score. Five samples were removed prior to MAGeCK analysis due to insufficient enrichment of control gRNAs, improper clustering, or poor library coverage (Suppl. Fig. 2).

**In situ T-tubule imaging**. CASAAV virus was injected subcutaneously at P1 into $R26^{Cas9/+}$;$Myh7^{YFP/+}$ pups. Animals were sacrificed at P28 and hearts were perfused with the T-tubule-binding dye FM 4-64 (Thermo, T3166) at 5 μM for 20 min, followed by direct imaging on an Olympus FV3000RS confocal microscope[5]. Organization and abundance of transverse and longitudinal T-tubule elements was quantified using AutoTT software[60].

**Immunostaining and phenotyping**. Freshly dissociated adult CMs were cultured on laminin-coated (2 μg/cm², Life Technologies, 23017015) 12-mm glass coverslips in a 24-well dish with DMEM plus 5% FBS and 10 μM blebbistatin (EMD Millipore, 203390) at 37 °C. After allowing cells to adhere to the coverslips for 30 min, CMs were fixed with 4% PFA for 10 min at room temperature, followed by permeabilization with PBST (PBS + 0.1% Triton) for 10 min at room temperature. Cells were then blocked with 4% BSA/PBS for 1 hr and incubated with primary antibody (Suppl. Table 4) diluted 1:500 in blocking solution overnight at 4 °C. The next day, CMs were briefly rinsed three times with 4% BSA/PBS and incubated with fluorescently conjugated secondary antibodies for 1 h at 4 °C. After rinsing three times with 4% BSA/PBS, coverslips were mounted on slides using Diamond Antifade mountant (Thermo, P36965), and imaged on an Olympus FV3000RS confocal microscope. For H2Bub1 staining, CMs were incubated in 1% SDS/PBS for 5 min, rinsed with PBS three times, and then blocked and stained as described above. Myh7YFP was visualized without staining for the YFP tag and percent YFP(+) was manually counted. Nucleation of each CM was also manually counted, while area, length, and width were measured using ImageJ. T-tubule organization was quantified using AutoTT software[60]. Cell size, dimensions, nucleation, and T-tubule organization were all measured on the same cells. In Fig. 2, to control for variation in dissociation quality, measurements were made on GFP+ (transduced) CMs and GFP− CMs (untransduced), and data from transduced CMs were normalized to that of untransduced CMs from the same mouse (Fig. 2). For histology, hearts were collected at P7 and fixed overnight in 4% PFA. Hearts were then transferred to 15% sucrose/PBS for 3 hr, and then to 30% sucrose overnight. Hearts were then cryoprotected in Tissue Freezing Medium (GeneralData Healthcare Cat.# TFM-COLOR) and stored at −80 °C. Hearts were then sectioned on a cryostat at 10 μm, and sections stored at −20 °C. Staining was performed as described above for dissociated CMs, except sections that were incubated in PBS + 0.1% Triton X-100 for 20 min to remove Tissue Freezing Medium before starting the staining protocol.

**Western blotting**. Ventricles were homogenized in 1 ml of RIPA buffer and agitated at 4 °C for 30 min. Lysates were spun at 10,000 g for 10 min at 4 °C, and the supernatant transferred to a new tube. About 40 μg of protein was boiled in SDS loading buffer for 5 min, loaded onto an Invitrogen Bolt 4–12% gradient precast mini gel (NW04120BOX), and run at 165 volts for 45 min. Protein was transferred to a preequilibrated PVDF membrane in transfer buffer (Boston BioProducts oBP-190) via Biorad Trans-Blot SD Semi-Dry Transfer Cell, at 20 volts for 40 min. Blots were cut and blocked in 2% milk/TBST for 1 h at 4 °C. Blots were then incubated overnight at 4 °C with primary antibodies (Suppl. Table 4) for H2Bub1, NPPA, and GAPDH, in block solution. The next day, blots were rinsed in TBST and incubated with HRP-conjugated donkey anti-rabbit secondary antibody (Rockland 611-703-127) diluted 1:20,000 in blocking solution for 1 h at room temperature. Blots were then rinsed in TBST, incubated in Millipore Immobilon Western Chemiluminescent HRP Substrate (WBKLS0500) for 1 minute, and imaged on a GE Healthcare ImageQuant LAS4000. In the case of RNF20/40, the blot was first probed for RNF20, then stripped, and reprobed for RNF40.

**RNA sequencing**. One day old $R26^{fsCas9-2A-GFP/+}$;$Myh7^{YFP/+}$ pups were subcutaneously injected with 50 μl of CASAAV containing the most enriched Rnf20-targeting gRNA and the most enriched Rnf40-targeting gRNA in a single vector (CASAAV-Rnf20/40) or a control vector containing Cre but no gRNAs. Vectors were injected at a concentration of $1 \times 10^{11}$ vg/ml, which was sufficient for ~20% CM transduction. At four weeks of age, mice were sacrificed and single-cell CM dissociations prepared by collagenase perfusion were subjected to FACS. YFP+ DKO CMs, or control GFP+ CMs (without regard for YFP), were sorted into Trizol

and RNA was extracted via standard phase separation. Approximately 100,000 CMs were collected from each of five DKO and five control mice. Trizol aqueous phase was transferred to a Zymo RNA Clean and Concentrator spin column and treated with DNase prior to elution in 20 μl of H₂O, as described above. Reverse transcription of 40 ng of RNA for each sample, and library amplification, was performed using the SMARTseq v4 Ultra Low Input RNA-seq kit (Clontech 634889). The standard manufacturer's protocol was followed, with eight cycles of amplification. To prepare amplified libraries for sequencing, 300 pg was fragmented and indexed using the Nextera XT DNA Library Preparation Kit (Illumina FC-131-1024) and Index Kit (Illumina FC-131-1001), according to the standard manufacturer's protocol. Single-end 75-bp sequencing of pooled libraries was performed on a NextSeq500. After trimming the first 15 bp from sample reads to remove adapter sequences, reads were aligned to the mm10 transcriptome (ftp://ftp.ensembl.org/pub/release-93/fasta/mus_musculus/cdna/) using Kallisto[61]. An average sequencing depth of ~32 M pseudoaligned reads per sample was achieved. Kallisto counts for individual transcripts were consolidated into gene-level counts using TxImport[62], and analyzed for differential expression between control and RNF20/40-depleted sample groups using DESeq2 (Suppl. Data 4)[59]. Gene ontology analysis of differentially expressed genes was conducted with Gene Set Enrichment Analysis[36], with 10,000 permutations and reported gene sets being limited to those containing at least 30 genes (Suppl. Table 2).

To compare gene expression between wild-type neonatal and mature CMs, we dissociated postnatal P1 or P28 heart ventricles. Neonatal CMs were isolated by Neonatal CM Isolation Kit (Cellutron, Cat# nc-6031) and purified using the Miltenyi Neonatal CM Isolation Kit (Miltenyi Biotec, 130-100-825). P28 CMs were dissociated by collagenase perfusion and purified by differential sedimentation. Total RNA was isolated from purified cells using the RNeasy mini kit (Qiagen), and rRNA was removed using the Ribo-Zero Gold kit (Illumina). RNA was converted into sequencing libraries using Script-seq v2 (Illumina). Single-end 75-nt sequencing was performed on a NextSeq500. We analyzed biological triplicate samples for each developmental stage.

Reads were mapped to mm10 using STAR[63], and FeatureCounts[64] was used to determine gene-level expression values. Differential expression between wild-type and P28 samples was analyzed using DESeq[59]. For construction of custom gene sets containing the genes with the highest fold change between stages, we ranked genes by their ratio between mature and neonatal CMs (Suppl. Data 5). The highest 100 genes were elected for an "adult specific gene set" and the lowest 100 for a "neonatal specific gene set" (Suppl. Table 1).

**ChIP sequencing**. Each final ChIP-seq sample consisted of 20 apexes from bisected P1 hearts, eight apexes from P7 hearts, or four apexes from P28 hearts. Each sample consisted of male and female tissue in equal proportions. Tissue, either two adult apexes, 4 P7 apexes, or one litter of P1 apexes, was harvested into ice cold PBS, transferred into 1.2 ml of 1% formaldehyde (Sigma F8775), and homogenized with a T10 Ultra Turrax homogenizer at setting six for 30 s. The homogenate was cross-linked at room temperature on a rotator for 25 min. Formaldehyde was quenched by adding glycine to a final concentration of 500 mM and incubating for 5 min at room temperature with rotation. The homogenate was then centrifuged at 3000 g for 3 min at 4 °C. The pellet was resuspended in 1 ml of cold PBS. This PBS wash was repeated a total of three times. After the third wash, the pellet was resuspended in hypotonic buffer (20 mM HEPES pH 7.5, 10 mM KCl, 1 mM EDTA, 0.1 mM activated Na₃VO₄, 0.5% NP-40, 10% glycerol, 1 mM DTT, and 1:1000 Roche cOmplete protease inhibitor), transferred to a glass dounce homogenizer, and dounced with pestle "B" (20x for P1, 100x for P7, or 200x for adult). Samples were transferred into siliconized Eppendorf tubes and incubated on ice for 15 min, then centrifuged at 13,000 g for 2 min at 4 °C. Pellets were stored in liquid nitrogen until multiple samples were ready for sonication.

When sufficient samples were available for ChIP, pellets were thawed, resuspended in 1 ml of hypotonic buffer per 20 neonatal apexes, four P7 apexes, or two adult apexes, and incubated on ice for 5 min. Lysate was transferred into glass dounce homogenizer and again dounced with pestle "B" as before. Lysate was centrifuged at 13,000 g for 2 min at 4 °C, and the pellet was resuspended in 500 μl of ChIP Dilution Buffer (20 mM Tris-Cl pH 8.0, 2 mM EDTA, 150 mM NaCl, and 1% Triton X-100) supplemented with SDS to 1%, and 1:50 protease inhibitor. Samples were then equally split into two 0.65-ml tubes (BrandTech 781310 or Corning 3208) for sonication. Samples were sonicated using a Qsonica 800 R, at 65% amplitude, 10 s on, 30 s off, for 25 min of on-time (100 min in total). After sonication, the half-samples were recombined in 1.5-ml siliconized tubes and centrifuged at 18,500 g for 5 min at 4 °C. The chromatin/supernatant was transferred to a new tube and a 20-μl aliquot was removed as an input sample for each replicate. In all, 120 μl of Protein A Dynabeads (Life Technologies 10002D) were prepared for each ChIP sample by washing beads in 1 ml of BSA solution (5%BSA/PBS) at 4 °C for 20 min. The wash was repeated for a total of three times, using a magnet to collect beads and remove BSA solution between washes. After the third wash, beads were resuspended in 1 ml of BSA solution, 250 μl of which was set aside to preclear the chromatin. In all, 5 μl of H2Bub1 antibody (Cell Signaling 5546), 3 μl of H3K4me3 antibody (Active Motif 39159), or 5 μl of H3K36me3 antibody (Cell Signaling 4909 S) was added to the remaining 750 μl of Dynabeads, along with 250 μl of fresh BSA solution, and incubated overnight at 4 °C. Sheared chromatin samples were

precleared by incubating with the 250-µl aliquot of Dynabeads resuspended in 2 ml of Dilution Buffer (20 mM Tris pH 8, 2 mM EDTA, 150 mM NaCl, 1% Triton X-100, and 1:50 Roche cOmplete protease inhibitor) for 1 h at 4 °C. Next, the antibody–Dynabead conjugate was rinsed three times in BSA solution, resuspended in 150 µl of Dilution Buffer, and added to the precleared chromatin. After 36 h of incubation at 4 °C, beads were collected on a magnet and washed three times with 1 ml of LiCl wash buffer (1% w/v Na-Deoxycholate, 500 mM LiCl, 1% NP40 Substitute, and 100 mM Tris, pH 7.5) and twice with TE buffer (10 mM Tris-HCl, 1 mM EDTA). After removing TE buffer, beads were suspended in 75 µl of SDS Elution Buffer (1% SDS, 0.1 M NaHCO$_3$). Samples were then incubated at 37 °C for 15 min with 1000-RPM mixing. This step was repeated once and the eluates were combined. To reverse crosslinking, input and immunoprecipitated samples were diluted to 200 µl with SDS Elution Buffer, NaCl was added to a final concentration of 200 mM, and Proteinase-K to a final concentration of 0.2 mg/ml. Samples were then incubated at 65 °C overnight. The next day, RNase-A was added to a final concentration of 1 mg/ml and incubated at 37 °C for 15 min. DNA was then purified using a Zymo DNA Clean and Concentrator column according to the manufacturer's instructions, and eluted in 100 µl of H$_2$O. NGS sequencing adapter addition and multiplex indexing of DNA samples was achieved using the KAPA Hyper Prep Kit (KAPA Biosystems KK8502) according to the manufacturer' instructions. About 25 ng of DNA was used as input. In all, 75-nt single-end sequencing was performed on a NextSeq500, or 150-bp paired end on an Illumina HiSeq. In cases where paired-end sequencing was performed, the reverse reads were discarded, and forward reads clipped to 75 bp, so that all sequencing could subsequently be processed in a uniform manner.

ChIP and input reads were aligned to mm10 using Bowtie2[65], and reads for each gene were extracted using Bedtools[66]. Profile plots were created using Deeptools[67]. For H2Bub1, library-size normalization was performed with DESeq2. Reads per gene were then normalized to gene length, from NCBI RefSeq first transcription start site to the last transcription stop site, to give a H2Bub1 binding score (Suppl. Data 6). Input values were subtracted from ChIP values for the final ChIP signal. Genes with ChIP signal >0 were considered to be occupied by H2Bub1. For H3K4me3 libraries were RPKM-normalized and reads at the TSS +/− 1 kb were scored using Bedtools (max signal per interval). Replicates were averaged and input signal was subtracted from ChIP signal. Only the highest-scoring TSS was used for each gene. To exclude genes with a low signal, a change in H3K4me3 was analyzed for genes with TSS scoring greater than 500 RPKM (~7000 genes). For H3K36me3 libraries were RPKM-normalized and reads per gene counted with Bedtools. Reads per gene were normalized to gene length, and ChIP replicates were averaged to get a final score for each gene. To exclude genes with very low signal, a change in H3K36me3 was analyzed for the top quintile (~7000 genes).

**Reporting summary**. Further information on research design is available in the Nature Research Reporting Summary linked to this article.

## Data availability

The data in this publication have been deposited in the GEO database under accession code GSE139975. Both raw and processed data are available. The raw data are also available on the Cardiovascular Development Consortium Server at https://b2b.hci.utah.edu/gnomex/, experiment number 483R (login as guest). The source data underlying bar, violin, and scatter plots in all figures and supplemental figures are provided in the Supplementary Information/Source Data file. The Riken Transcription Factor Database (TFdb) can be accessed at http://genome.gsc.riken.jp/TFdb/htdocs/. The Animal Transcription Factor Database (AnimalTFDB) can be accessed at http://bioinfo.life.hust.edu.cn/AnimalTFDB/#!/. Single cell cardiac RNA-seq data were mined from Panglao database (PanglaoDB), accession code SRA653146 (https://panglaodb.se/view_data.php?sra=SRA653146&srs=SRS2874273). Source data are provided with this paper.

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

## Acknowledgements

The authors thank the Dana-Farber Flow Cytometry Core for assistance with cell sorting and Eswar Prasad for guidance related to gRNA reverse transcription. N.J.V. was supported by NIH (T32HL007572, F32HL13423501, K99HL143194) and the Boston Children's Hospital Kaplan Fellowship. W.W.P. was supported by NIH (2UM1 HL098166, R01 HL146634) and the American Heart Association (17IRG33410894). The content is solely the responsibility of the authors and does not necessarily represent the official views of the National Heart, Lung, and Blood Institute or the National Institutes of Health. Portions of this research were conducted on the O2 High Performance Compute Cluster, supported by the Research Computing Group, at Harvard Medical School. See http://rc.hms.harvard.edu for more information.

## Author contributions

N.J.V. and W.W.P. conceived and designed the study. N.J.V. executed most experiments. J.Y.L., W.G., C.E.B., Y.Z. and J.S.K. generated plasmids, viruses, and other necessary reagents, and assisted with processing tissues and cells. Y.G. assisted with experimental design. P.Z. conducted wild-type C.M.-specific P0 and 4-wk RNA sequencing. Q.M. conducted echocardiography. N.J.V., I.S., S.S., G.C.Y. and W.W.P. analyzed the data. N.J.V. and W.W.P. wrote the paper.

## Competing interests

The authors declare no competing interests.
