## [Peer Review File · Nature Communications]

Reviewers' comments:

Reviewer #1 (Remarks to the Author):

This paper by VanDusen et al is interesting and important on two levels. First, it reports an innovative forward genetic screening method that should be widely applicable to define factors required for specific biological pathways or states. Second, using that screen, the authors defined a set of transcription related proteins required for cardiomyocyte (CM) maturation in the mouse, including the RNF20/40 ubiquitin ligase complex. The screen is well designed and well controlled, with positive and negative controls. RNF20/40 are best characterized in terms of monoubiquitination of histone H2B, which is associated with gene transcription. As such, the authors relate changes in gene expression (by RNA seq) upon depletion of RNF20/40 with changes in H2Bub1 (by ChIP-seq), and they identify a set of genes that are functionally connected to CM maturation. H2Bub1 is a complicated epigenetic mark, as it is regulated by multiple factors, including deubiquitinases, and cycles of H2B ub and de-ub are important for transcription. Use of this dynamic mark as a proxy for RNF20/40 dependent genes is therefore not perfect. Addition of ChIP data for RNF40 or RNF20, or for other histone modifications (H3K4me3) associated with gene activation in RNF40/20 depleted cells would strengthen the conclusions. Also, RNF20/40 have other, nonhistone substrates, so the authors should temper their statements regarding cause and effect, i.e. on on page 7 " RNF20/40 as the first epigenetic regulators of CM maturation being required to deposit H2B1ub1"---you cannot be 100% certain the effects are due to H2bub1; also, RNF20/40 do not deposit histones, they modify them. Similarly on page 8 "disrupting H2Bub1 deposition, which prevented normal transcriptional changes" is a bit of an overstatement.

Reviewer #2 (Remarks to the Author):

The manuscript is of high quality, both interesting and well performed. I have the following questions:
Major:

- The authors investigate RNF20/40 function through depletion in neonatal cardiomyocytes. Do expression and function of RNF20/40 change during cardiac development, also compared to adulthood?
- The authors identify RNF20/40 targets by correlating gene expression changes with alterations of K120 ubiquitination of histone H2B in mice depleted for the two epigenetic genes. Is there a particular reason why ChIP or ChIP seq experiments were not performed? This could be a valuable experiment to determine whether the modification of the expression of selected genes depends upon RNF20/40 activity.
- Since a cross-talk exists between H2Bub1 and other histone modifications and since H3K36me3 is also involved in transcription elongation, is there a consequential effect on H3K36me3 after RNF20/40 depletion?

Minor comments:

- Please specify what kind of analysis has been conducted in figure 4f? It is unclear (Kegg pathway gene set enrichment analysis or biological processes)
- In figure 4f the distribution between genes up and down is not clear. Please make it clearer.

Reviewer #3 (Remarks to the Author):

This manuscript by VanDusen et al. is a nice extension of the previous work performed by the same group (Guo and VanDusen et al., *Circulation Research* 2017), which aims to identify factors regulating CM maturation. A Myh7-YFP mouse model was used for an in vivo CRISPR-based screen, which allowed single-cell, fluorescent-based readout that can be analyzed with FACS coupled with next generation sequencing. RNF20/40, writers for H2Bub1 modification, were identified and characterized. The experiments were well-designed and performed. The study provides new insights on CM maturation, and demonstrates the feasibility and power of in vivo CRISPR screen. Overall, I think it has potentials to be published in *Nature Communications*. I have a few minor comments.

1. How many YFP+ cells were sorted for sequencing? It's difficult to appreciate the number of CMs analyzed from Suppl. Fig. 2a. Units for axes are missing in the plots. In addition, please provide original FACS plots to show how gating on YFP and GFP was determined.
2. One potential issue with pooled AAV vector cloning and virus synthesis is the uncontrollable representation of each vector in the final viral particles, potentially due to random recombination of AAV vectors during the production. Thus it is important to show in supplemental data how many out of the 14675 sgRNAs were consistently detected across all samples and their relative proportion in the entire library. Please also provide a quantitative representation of each sgRNAs in the inputs.
3. It was unclear how the data shown in Fig. 2c-f was done. Were these measurements done on dissociated heart cells or on fixed heart sections? Please include method descriptions for these analyses.
4. In Figure 4g, authors should label the genes that were either downregulated or upregulated by CASAAV-RNF20/40. This will help to draw correlation between RNF20/40 regulated genes and the maturation-associated genes shown on the graph.
5. Please discuss why some candidates from top 10 screen hits (*Poc1b*, *Setd6*, *Eif3l*) did not show persistent YFP expression (Fig.2a).
6. The authors are encouraged to discuss on the potential mechanisms that regulate RNF20/40 expression during CM maturation. Are they differentially expressed when comparing P1 vs P28 ventricles? Since Pu lab has performed an impressive characterization of cardiac TFs in Akerberg and Gu et al., *Nature Communications* 2019, can RNF20/40 and other hits identified from this study be potentially regulated by key cardiac TFs, which they have previously analyzed using bioChIP?
7. In Supp Fig 4, please show cardiac histology given the variability in echo data. Aside from gross cardiac morphology, please also demonstrate the difference in mono vs binucleation in vivo.

Reviewers' comments:

Reviewer #1 (Remarks to the Author):

This paper by VanDusen et al is interesting and important on two levels. First, it reports an innovative forward genetic screening method that should be widely applicable to define factors required for specific biological pathways or states. Second, using that screen, the authors defined a set of transcription related proteins required for cardiomyocyte (CM) maturation in the mouse, including the RNF20/40 ubiquitin ligase complex. The screen is well designed and well controlled, with positive and negative controls.

We appreciate the reviewer's positive comments.

RNF20/40 are best characterized in terms of monoubiquitination of histone H2B, which is associated with gene transcription. As such, the authors relate changes in gene expression (by RNA seq) upon depletion of RNF20/40 with changes in H2Bub1 (by ChIP-seq), and they identify a set of genes that are functionally connected to CM maturation. H2Bub1 is a complicated epigenetic mark, as it is regulated by multiple factors, including deubiquitinases, and cycles of H2B ub and de-ub are important for transcription. Use of this dynamic mark as a proxy for RNF20/40 dependent genes is therefore not perfect. Addition of ChIP data for RNF40 or RNF20, or for other histone modifications (H3K4me3) associated with gene activation in RNF40/20 depleted cells would strengthen the conclusions.

RNF20/40 ChIP-seq data was not included in our original study due to a lack of ChIP-grade antibodies. To our knowledge successful ChIP-sequencing of RNF20 or RNF40 has not been reported, and cardiac tissue is particularly challenging. During the revision period we attempted to circumvent antibody limitations by cardiomyocyte-specific overexpression of epitope tagged RNF20 and RNF40. Despite observing successful overexpression and nuclear localization of HA-tagged RNF20 and Myc-tagged RNF40, neither factor was successfully ChIP sequenced (figure below). As RNF20/40 do not directly bind DNA, and their interactions with H2B may be quite transient, this result is not entirely surprising. We hope the reviewers will be understanding of the technical limitations that prevent us from generating these data.

Unsuccessful ChIP-seq of RNF20 and RNF40. **a.** AAV9 was used to overexpress HA-tagged RNF20, along with GFP, in neonatal CMs. **b.** Immunostaining for the HA-tag in isolated adult CMs showed proper localization of the overexpressed RNF20 to the nucleus. **c-d.** Quantification of overexpression by capillary western blot of P7 whole apex tissue showed approximately 2.25 fold overexpression. **e.** AAV9 was used to overexpress Myc-tagged RNF40, along with mCherry, in neonatal CMs. **f.** Myc-tagged RNF40 localized to the nucleus. **g-h.** Capillary western blot showed approximately 2.5 fold overexpression of RNF40. **i.** Apex tissue from P7 hearts overexpressing epitope tagged RNF20 or RNF40 was used for ChIP-seq. However, ChIP-signal for both factors was diffuse and failed to differentiate from input control samples.

To better characterize the relationship between H2Bub1 and additional histone modifications, we conducted H3K4me3 and H3K36me3 ChIP-seq in H2Bub1 deficient hearts. To generate hearts with more complete and uniform H2Bub1 depletion in cardiomyocytes than possible with somatic Cas9 mutagenesis, we acquired an *Rnf20* flox allele. We confirmed that the *Rnf20* flox allele mosaic knockout single cell phenotype is comparable to the CASA- based RNF20/40 double knockout (RNF Cas-KO). These data validate the CASA- based phenotypic

characterization and are provided as a new supplemental figure (Suppl. Fig. 5). We then conducted H3K4me3 and H3K36me3 ChIP-sequencing in P7 RNF20^{flox/flox} or wild type control hearts that had been transduced with a high dose of Cre recombinase at P0. We showed that a subset of genes had differential marking of promoter regions by H3K4me3, or of gene bodies by H3K36me3. Next we used gene set enrichment analysis to show that the differentially marked genes were enriched for genes downregulated in RNF Cas-KO (Fig. 6c,d lower panels), genes upregulated during maturation (Suppl. Fig. 8a,b, gene set “Matur. up”), and genes that are upregulated during maturation, gain H2Bub1 during maturation, and are downregulated in the RNF Cas-KO (Suppl. Fig. 8a,b, gene set “Intersect. mature”). The mature sarcomere isoforms *Myh6* and *Tnni3*, which were downregulated in the RNF Cas-KO, had reduced occupancy by H3K4me3 and H3K36me3 upon *Rnf20* KO. Overall, the correlation between change in gene expression in the RNF Cas-KO and change in H3K4me3 or H3K36me3 occupancy was poor (Suppl. Fig. 8c,d), suggesting that changes in occupancy were not simple downstream effects of altered gene expression levels. These data have been added as a new figure (Fig. 6), and a new supplemental figure (Suppl. Fig. 8). Description and discussion have been added to the text.

Also, RNF20/40 have other, nonhistone substrates, so the authors should temper their statements regarding cause and effect, i.e. on page 7 “RNF20/40 as the first epigenetic regulators of CM maturation being required to deposit H2B1ub1”---you cannot be 100% certain the effects are due to H2bub1; also, RNF20/40 do not deposit histones, they modify them. Similarly on page 8 “disrupting H2Bub1 deposition, which prevented normal transcriptional changes” is a bit of an overstatement.

The reviewer is correct that a number of non-histone substrates have been identified and we cannot be certain that these substrates don't affect the maturational phenotypes that we report. We acknowledge this possibility in the revised text:

“Depletion of *Rnf20/40* broadly impaired CM maturation by preventing essential transcriptional changes. H2B is the primary target of RNF20/40-mediated monoubiquitination, and is the most likely mediator of the phenotypes that we observed, however, non-histone RNF20/40 substrates have been identified(1–4), and it is possible that they also contribute to gene regulation of CM maturation.”

We also clarify our language regarding H2B modification by monoubiquitination, referring to RNF20/40 writing of H2Bub1 rather than deposition.

Reviewer #2 (Remarks to the Author):

The manuscript is of high quality, both interesting and well performed. I have the following questions:

Major:

- The authors investigate RNF20/40 function through depletion in neonatal cardiomyocytes. Do expression and function of RNF20/40 change during cardiac development, also compared to adulthood?

We agree that these are important questions. Somatic Cas9 and AAV-mediated mutagenesis (CASA-AV) is not ideal for adult stage knockouts, as CMs become polyploid during the maturation process, which makes simultaneous knockout of all alleles via indel formation difficult. To circumvent this difficulty we acquired an *Rnf20* flox allele. We generated mosaic adult knockout of *Rnf20* by administering AAV9-Cre at 2 months of age and assessed the impact on cell morphology at 3 months. Knockout CMs showed no difference in length, width, or area, while T-tubule organization was disrupted. These data indicate that RNF20 is required for some aspects of adult stage homeostasis, such as maintenance of the T-tubule network, but is not required for others. We now include these data, along with the neonatal stage *Rnf20* flox allele based KO, as a new supplemental figure (Suppl. Fig. 5).

Page 7-8: "As CMs mature they become polyploid, making simultaneous mutagenesis of all alleles by CASA-AV-based indel formation more difficult. To bypass this difficulty, and further validate our CASA-AV-based findings, we acquired a conditional *Rnf20* allele, *Rnf20^{fx}*. Injection of *Rnf20^{fx}*; R26^{fs}Tomato newborn pups with a mosaic dose of AAV-TnT-Cre resulted in robust depletion of H2Bub1 within transduced CMs (RNF20-KO; Suppl. Fig. 5a,b). Subsequent analysis of CM area, length, width, and T-tubule organization revealed an immature phenotype consistent with our CASA-AV-based observations (Suppl. Fig. 5c-i). To determine if the dependence of mature CM phenotype on RNF20 is stage specific, we treated *Rnf20^{flox}*;R26^{fs}Tomato mice with a mosaic dose of AAV-TnT-Cre at 8 weeks of age and dissociated CMs for analysis at 12 weeks. Area, length, and width of CMs was unaffected by mosaic RNF20 KO (Suppl. Fig. 5j-l). In contrast, T-tubule organization was disrupted, with KO CMs demonstrating decreased transverse element density and increased longitudinal element density (Suppl. Fig. 5m-p). These data indicate that RNF20 is necessary in adult CMs for some maintenance of some features of maturation (T-tubules) but not others (size and overall morphology)."

To address developmental expression of RNF20/40 we have also included a new supplemental figure (Suppl. Fig. 4) that shows the distribution of RNF20/40 expression within different cardiac cell populations (mined from a publicly available single cell RNA-seq dataset), the RNA expression of each factor at neonatal and adult timepoints in cardiomyocytes, and the ChIP-seq binding profiles of core cardiac transcription factors at the RNF20 and RNF40 loci (mined from previously reported Pu lab datasets). In short, RNF20/40 are expressed in several cardiac cell populations, and within cardiomyocytes their expression level is relatively stable during development. This expression is likely driven by core cardiac transcription factors, many of which robustly bind at the promoter for each factor. RNF20/40 activity may be regulated post-transcriptionally, through developmentally regulated co-factor interaction, or through developmentally regulated recruitment to target loci.

Page 7: “These results could suggest developmental post-transcriptional regulation of RNF20/40 function, developmental regulation of co-factors, or developmental regulation of RNF20/40 recruitment to regulate genes.”

- The authors identify RNF20/40 targets by correlating gene expression changes with alterations of K120 ubiquitination of histone H2B in mice depleted for the two epigenetic genes. Is there a particular reason why ChIP or ChIP seq experiments were not performed? This could be a valuable experiment to determine whether the modification of the expression of selected genes depends upon RNF20/40 activity.

See response to reviewer one. In short, ChIP grade antibodies for RNF20/40 are not available, and our attempts to ChIP-sequence overexpressed epitope-tagged RNF factors were unsuccessful. To our knowledge, RNF20/40 ChIP-seq has not been reported previously suggesting that this challenge has also been shared by other groups studying this epigenetic regulator.

- Since a cross-talk exists between H2Bub1 and other histone modifications and since H3K36me3 is also involved in transcription elongation, is there a consequential effect on H3K36me3 after RNF20/40 depletion?

We conducted H3K4me3 and H3K36me3 ChIP-seq in H2Bub1 deficient hearts. To generate hearts with more complete and uniform H2Bub1 depletion in cardiomyocytes than possible with somatic Cas9 mutagenesis, we acquired an *Rnf20* flox allele. We confirmed that the *Rnf20* flox allele mosaic knockout single cell phenotype is comparable to the CASA AV-based RNF20/40 double knockout (RNF Cas-KO). These data validate the CASA AV-based phenotypic characterization and are provided as a new supplemental figure (Suppl. Fig. 5). We then conducted H3K4me3 and H3K36me3 ChIP-sequencing in P7 RNF20^{flox/flox} or wild type control hearts that had been transduced with a high dose of Cre recombinase at P0. We showed that a subset of genes had differential marking of promoter regions by H3K4me3, or of gene bodies by H3K36me3. Next we used gene set enrichment analysis to show that the differentially marked genes were enriched for genes downregulated in RNF Cas-KO (Fig. 6c,d lower panels), genes upregulated during maturation (Suppl. Fig. 8a,b, gene set “Matur. up”), and genes that are upregulated during maturation, gain H2Bub1 during maturation, and are downregulated in the RNF Cas-KO (Suppl. Fig. 8a,b, gene set “Intersect. mature”). The mature sarcomere isoforms *Myh6* and *Tnni3*, which were downregulated in the RNF Cas-KO, had reduced occupancy by H3K4me3 and H3K36me3 upon *Rnf20* KO. Overall, the correlation between change in gene expression in the RNF Cas-KO and change in H3K4me3 or H3K36me3 occupancy was poor (Suppl. Fig. 8c,d), suggesting that changes in occupancy were not simple downstream effects of altered gene expression levels. These data have been added as a new figure (Fig. 6), and a new supplemental figure (Suppl. Fig. 8). Description and discussion have been added to the text.

Minor comments:

- Please specify what kind of analysis has been conducted in figure 4f? It is unclear (Kegg pathway gene set enrichment analysis or biological processes)

Figure 4f was a plot of change in RNA expression versus change in H2Bub1 ChIP signal. Perhaps the reviewer meant 4h (now figure 5h), which is an Ingenuity Pathway Analysis for enriched terms. We have updated the figure legend to include this information.

- In figure 4f the distribution between genes up and down is not clear. Please make it clearer.

To make the difference in distribution more clear we have added box plots next to the scatter plot. The plot was moved to Suppl. Fig. 7e.

Reviewer #3 (Remarks to the Author):

This manuscript by VanDusen et al. is a nice extension of the previous work performed by the same group (Guo and VanDusen et al., *Circulation Research* 2017), which aims to identify factors regulating CM maturation. A Myh7-YFP mouse model was used for an in vivo CRISPR-based screen, which allowed single-cell, fluorescent-based readout that can be analyzed with FACS coupled with next generation sequencing. RNF20/40, writers for H2Bub1 modification, were identified and characterized. The experiments were well-designed and performed. The study provides new insights on CM maturation, and demonstrates the feasibility and power of in vivo CRISPR screen. Overall, I think it has potentials to be published in *Nature Communications*. I have a few minor comments.

1. How many YFP+ cells were sorted for sequencing? It's difficult to appreciate the number of CMs analyzed from Suppl. Fig. 2a. Units for axes are missing in the plots. In addition, please provide original FACS plots to show how gating on YFP and GFP was determined.

In total approximately 365,000 YFP+ CMs were collected from 4M transduced CMs. The plots in Suppl. Fig. 2a are saved FACS plots from screen and control samples. We added units to the axes as requested. The GATA4/6 double KO positive control and Cre-only negative control were used to set the YFP gating of GFP+ CMs. An example of typical GFP gating is shown in revised Fig. 3f. These notes have been added to the methods section.

2. One potential issue with pooled AAV vector cloning and virus synthesis is the uncontrollable representation of each vector in the final viral particles, potentially due to random recombination of AAV vectors during the production. Thus it is important to show in supplemental data how many out of the 14675 sgRNAs were consistently detected across all samples and their relative proportion in the entire library. Please also provide a quantitative representation of each sgRNAs in the inputs.

The gRNAs in the library were well covered in the AAV library. We have added these data to Suppl. Fig. 2g-i.

3. It was unclear how the data shown in Fig. 2c-f was done. Were these measurements done on dissociated heart cells or on fixed heart sections? Please include method descriptions for these analyses.

These measurements were done on dissociated immunostained CMs. We have indicated as such in the figure legend, and have added more details of the analysis to the methods section (Immunostaining and Phenotyping).

4. In Figure 4g, authors should label the genes that were either downregulated or upregulated by CASAAV-RNF20/40. This will help to draw correlation between RNF20/40 regulated genes and the maturation-associated genes shown on the graph.

In revised Fig. 5f, the genes that are differentially expressed in the RNF20/40 KO are labeled in red (upregulated) or blue (downregulated).

5. Please discuss why some candidates from top 10 screen hits (*Poc1b*, *Setd6*, *Eif3l*) did not show persistent YFP expression (Fig.2a).

Setd6 and *Eif3l* did modestly increase YFP expression, and this may have led to their positive score. *Poc1b* did not detectably increase YFP expression. Its ranking among the top 10 candidates may represent a false discovery. Alternatively, since most cells were transduced by multiple guides in random combinations, it is possible that loss of this gene sensitizes cells so that they are more likely to activate YFP in combination with other gRNAs. This could allow the gRNA to score highly in the screen but fail to activate YFP on its own. Notably, all three of the gRNAs that did not validate had less than three gRNAs with enrichment p-value < 0.001. This suggests that requiring at least 3 gRNAs to have significant enrichment could reduce false positives that fail to validate.

Page 5-6: "Of the three candidates that did not validate, one (*Poc1b*) was a false positive as it did not increase YFP expression. The remaining two (*Setd6* and *Eif3l*) caused modest but significant persistence of YFP. Notably, all three of these candidates had less than three gRNAs with DESeq2 enrichment p-value <0.001, suggesting an additional criteria that could improve screen stringency."

6. The authors are encouraged to discuss on the potential mechanisms that regulate RNF20/40 expression during CM maturation. Are they differentially expressed when comparing P1 vs P28 ventricles? Since Pu lab has performed an impressive characterization of cardiac TFs in Akerberg and Gu et al., Nature Communications 2019, can RNF20/40 and other hits identified from this study be potentially regulated by key cardiac TFs, which they have previously analyzed using bioChIP?

To address developmental expression of RNF20/40 we have also included a new supplemental figure (Suppl. Fig. 4) that shows the distribution of RNF20/40 expression within different cardiac cell populations (mined from a publicly available single cell RNA-seq dataset), the RNA expression of each factor at neonatal and adult timepoints in cardiomyocytes, and the ChIP-seq binding profiles of core cardiac transcription factors at the RNF20 and RNF40 loci (mined from previously reported Pu lab datasets). In short, RNF20/40 are expressed in several cardiac cell populations, and within cardiomyocytes their expression level is relatively stable during development. This expression is likely driven by core cardiac transcription factors, many of which robustly bind at the promoter for each factor. RNF20/40 activity may be regulated post-

transcriptionally, through developmentally regulated co-factor interaction, or through developmentally regulated recruitment to target loci.

Page 7: “These results could suggest developmental post-transcriptional regulation of RNF20/40 function, developmental regulation of co-factors, or developmental regulation of RNF20/40 recruitment to regulate genes.”

7. In Supp Fig 4, please show cardiac histology given the variability in echo data. Aside from gross cardiac morphology, please also demonstrate the difference in mono vs binucleation in vivo.

The requested histological data has been added (Fig. 3e). While we didn't observe severe morphological defects at P7, the high and widespread expression of Myh7 suggests substantial cardiac stress. The variability in echo data is likely due to acute onset of dysfunction, with some pups still having normal function at P7, but rapidly declining within a few days (Fig 3c). We haven't quantified nucleation because the early time point these hearts were collected at (P7) may precede karyokinesis in many CMs, and because this is a high dose knockout with organ-level dysfunction that will confound analysis of maturation related phenotypes. Cardiac dysfunction is a known stimulator of polyploidy.

REFERENCES

1. Duan Y, Huo D, Gao J, Wu H, Ye Z, Liu Z, Zhang K, Shan L, Zhou X, Wang Y, Su D, Ding X, Shi L, Wang Y, Shang Y, Xuan C. Ubiquitin ligase RNF20/40 facilitates spindle assembly and promotes breast carcinogenesis through stabilizing motor protein Eg5. *Nat Commun.* 2016 Aug 25;7:12648.
2. In S, Lee Y, Lee J, Kim J. Identification and characterization of eEF1B δ L as a substrate of RNF20/40. *Conference of the Korean Society for ...* [Internet]. 2017; Available from: <https://koasas.kaist.ac.kr/handle/10203/237407>
3. Liu Z, Oh S-M, Okada M, Liu X, Cheng D, Peng J, Brat DJ, Sun S-Y, Zhou W, Gu W, Ye K. Human BRE1 is an E3 ubiquitin ligase for Ebp1 tumor suppressor. *Mol Biol Cell.* 2009 Feb;20(3):757–68.
4. Chin L-S, Vavalle JP, Li L. Staring, a novel E3 ubiquitin-protein ligase that targets syntaxin 1 for degradation. *J Biol Chem.* 2002 Sep 20;277(38):35071–9.

REVIEWERS' COMMENTS

Reviewer #1 (Remarks to the Author):

This revised paper is even stronger than the original submission. The authors have carefully addressed the reviewers' questions and suggestions. The addition of new ChIP-seq data for H3K4me and K36me further defines changes in the epigenetic landscape impacted by RNF20/40 loss. This reviewer is understands well the difficulties in ChIPs for factors that interact transiently with chromatin, so absence of direct ChIP for RNF20 or RNF40 is not viewed as a weakness, but simply as the state of the field. This paper will be of interest to all studying CM maturation and to those studying RNF20/40. In addition, the screen methodology will be widely useful.

Reviewer #2 (Remarks to the Author):

The authors have replied to my criticisms thoroughly. I have no further questions. The manuscript is of high level and informative.

Reviewer #3 (Remarks to the Author):

The paper is acceptable for publication.